# SPEC-RL: ACCELERATING ON-POLICY REINFORCEMENT LEARNING VIA SPECULATIVE ROLLOUTS

## ABSTRACT

Large Language Models (LLMs) increasingly rely on reinforcement learning with verifiable rewards (RLVR) to elicit reliable chain-of-thought reasoning. However, the training process remains bottlenecked by the computationally expensive rollout stage. Existing acceleration methods—such as parallelization, objective- and data-driven modifications, and replay buffers—either incur diminishing returns, introduce bias, or overlook redundancy across iterations. We identify that rollouts from consecutive training epochs frequently share a large portion of overlapping segments, wasting computation. To address this, we propose **SPEC-RL**, a novel framework that integrates **SPEC**ulative decoding with the **RL** rollout process. SPEC-RL reuses prior trajectory segments as speculative prefixes and extends them via a draft-and-verify mechanism, avoiding redundant generation while ensuring policy consistency. Experiments on diverse math reasoning and generalization benchmarks, including GSM8K, MATH-500, OlympiadBench, MMLU-STEM, and others, demonstrate that SPEC-RL reduces rollout time by 2–3× without compromising policy quality. As a purely rollout-stage enhancement, SPEC-RL integrates seamlessly with mainstream algorithms (e.g., PPO, GRPO, DAPO), offering a general and practical path to scale RLVR for large reasoning models.

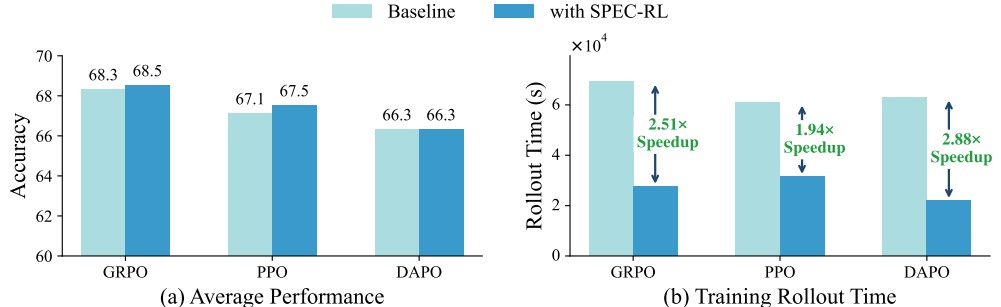

Figure 1: SPEC-RL achieves a 2–3× reduction in rollout time while maintaining average performance on Qwen-3-8B-Base across different algorithms.

## 1 INTRODUCTION

Large Language Models (LLMs) have recently achieved substantial progress on challenging reasoning-intensive tasks, such as mathematical problem-solving (Lewkowycz et al., 2022b), program synthesis (Chen et al., 2021; Li et al., 2022), and multi-step agentic planning (Yao et al., 2023b;a). A key enabler of these advances is reinforcement learning with verifiable rewards (RLVR) (Lambert et al., 2024; Guo et al., 2025; Yue et al., 2025), which has emerged as a widely adopted paradigm for incentivizing models to produce faithful and reliable chain-of-thought (CoT) reasoning (Wei et al., 2022). However, RLVR training pipelines remain constrained by the rollout stage, a fundamental efficiency bottleneck, despite its demonstrated efficacy (Zheng et al., 2025). During this stage, the model must generate large quantities of trajectories through interaction with the environment, a process that is computationally expensive and scales poorly with model size.

As a result, the cost and latency of trajectory generation dominate overall training time, severely limiting the practicality of scaling RLVR to increasingly capable LLMs.

To mitigate rollout inefficiency, prior work has explored three directions. First, parallelized rollout generation increases throughput by producing many trajectories per iteration (Xu et al., 2025), but its benefits fade as computational and synchronization costs rise. Second, model-based accelerations reduce environment interaction through modified objectives (Brantley et al., 2025; Lin et al., 2025), data restructuring (Liu et al., 2025; Zhang et al., 2025b), or sample selection heuristics (Yu et al., 2025; Zheng et al., 2025), though these approaches often introduce bias and added complexity. Third, caching methods such as replay buffers reuse prior trajectories (Zhang et al., 2025a), thereby improving data utilization, but still require fresh on-policy rollouts and struggle when policies shift significantly.

In this paper, we identify a key opportunity through a preliminary study that measures the token overlap ratio between consecutive epochs, using ROUGE-1 (Lin, 2004), across different algorithms (GRPO, PPO, and DAPO). We find that the overlap is already substantial from the second epoch (around 0.5) and gradually increases as training progresses, stabilizing around 0.7 in later epochs (Figure 2). This indicates that a substantial portion of sampled trajectories is repeatedly regenerated across training rounds, reflecting a strong potential for reducing rollout cost. Such redundancy naturally arises due to incremental policy updates, with the current policy often behaving similarly to the previous one. Moreover, in environments with fixed initial states or tasks (e.g., repeated prompts in an LLM reasoning task), the early parts of trajectories tend to overlap across iterations. As a result, significant computation is wasted regenerating these overlapping segments. This motivates the central question of our work: *can such redundancy be systematically exploited to accelerate rollouts?*

We answer this question by proposing **SPEC-RL**, a novel framework that integrates **SPEC**ulative decoding with the **RL** rollout process. Rather than regenerating full trajectories from scratch, SPEC-RL treats old rollouts from the previous epoch as implicit drafts: following the speculative decoding paradigm, old rollout tokens are verified under the current policy to form a verified prefix. When the first rejection position is reached, the current policy continues generation from that point onward, as illustrated in Figure 3. This approach is directly analogous to draft-and-verify methods in text generation, where a draft sequence is proposed and then validated in parallel by the target model (Leviathan et al., 2023). By incorporating the same mechanism into RL rollouts, SPEC-RL leverages cached rollouts to skip redundant computation while ensuring that the final outputs remain faithful to the current policy. The verified prefix is quickly extended by the latest policy, ensuring that the final trajectory remains consistent with the current policy's behavior.

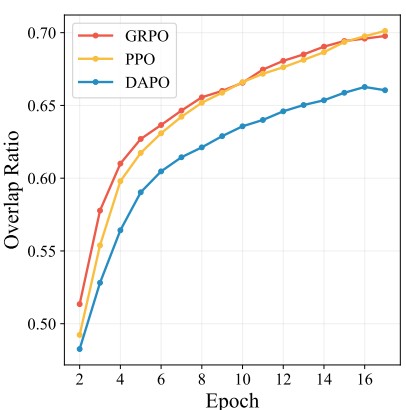

Figure 2: Token overlap ratio per epoch under GRPO, PPO, and DAPO. We compute the ratio using ROUGE-1, comparing rollout response tokens from each epoch against those from the previous consecutive epoch.

Our experiments demonstrate that SPEC-RL substantially improves training efficiency across diverse tasks and model scales. Concretely, SPEC-RL consistently reduces rollout generation time by 2–3× on average, while maintaining or even improving final policy performance across a wide range of math reasoning benchmarks (GSM8K (Cobbe et al., 2021), MATH-500 (Hendrycks et al., 2021), Minerva Math (Lewkowycz et al., 2022a), OlympiadBench (He et al., 2024), AMC 2023 (Art of Problem Solving, 2024)) and out-of-distribution benchmarks (MMLU-STEM (Hendrycks et al., 2020), IFEval (Zhou et al., 2023)). Importantly, SPEC-RL is designed as a modular enhancement to the data collection phase, making it readily applicable to a wide range of mainstream RLVR algorithms, such as GRPO, DAPO, and PPO.

In summary, we identify substantial rollout redundancy in RLVR and show that reusing overlapping trajectory segments can greatly reduce sampling cost. Building on this insight, we introduce SPEC-RL, the first framework to incorporate speculative decoding into RL rollouts by treating previous-

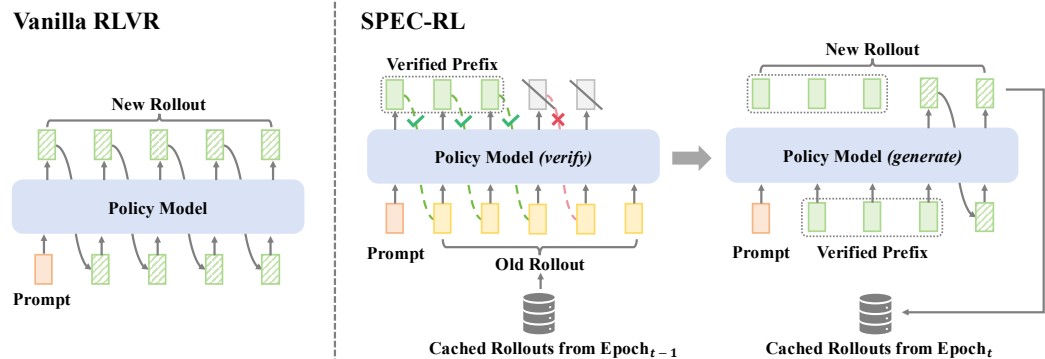

Figure 3: Comparison of the rollout process in Vanilla RLVR and SPEC-RL. Vanilla RLVR regenerates full responses at each epoch. In SPEC-RL, at each epoch $t = 1, \ldots, T$, cached rollouts from the previous epoch are verified in parallel to retain verified prefixes, the remaining tokens are discarded, and generation resumes from the rejection position before assembling the final response.

epoch trajectories as implicit drafts and verifying reusable prefixes. This design integrates smoothly with mainstream RL algorithms and yields significant rollout acceleration while maintaining policy performance.

## 2 PRELIMINARIES

### 2.1 ON-POLICY REINFORCEMENT LEARNING

Reinforcement learning with verifiable rewards (RLVR) formulates the answer generation of an LLM as a conditional sampling policy. Given a dataset of reasoning pairs $(\mathbf{x}, \mathbf{y}^*) \sim \mathcal{D}$, where $\mathbf{x}$ is a prompt and $\mathbf{y}^*$ is the ground-truth answer, the policy $\pi_\theta(\cdot \mid \mathbf{x})$ generates a candidate response $\mathbf{y}$. A reward function $R(\mathbf{y}, \mathbf{y}^*)$ evaluates whether the generated response $\mathbf{y}$ matches the ground-truth answer $\mathbf{y}^*$. Training relies on on-policy rollouts, where samples are drawn from the current policy at every iteration. This ensures that training data remain consistent with the current policy distribution, avoiding the distribution mismatch issues common in off-policy methods and yielding more stable learning. However, the downside is that new trajectories must be regenerated at each update, and the cost of producing long sequences makes rollout the dominant efficiency bottleneck in RLVR. The objective of on-policy RL is simply to maximize the expected reward of the generated responses:

$$J(\theta) = \mathbb{E}_{(\mathbf{x}, \mathbf{y}^*) \sim \mathcal{D}, \mathbf{y} \sim \pi_\theta(\cdot \mid \mathbf{x})} \big[ R(\mathbf{y}, \mathbf{y}^*) \big]. \tag{1}$$

In this work, we keep the RL objective and policy update unchanged and focus on improving the efficiency of the rollout stage.

### 2.2 SPECULATIVE DECODING

Speculative decoding follows a draft–and–verify paradigm: an efficient draft model $p$ (e.g., a smaller LM) first drafts multiple future tokens, and the target model $q$ verifies them in parallel. A drafted token $z_i \sim p(\cdot \mid \mathbf{x}, \mathbf{z}_{<i})$ is accepted with

$$\alpha(z_i \mid \mathbf{x}, \mathbf{z}_{<i}) = \min\left\{ 1, \frac{q(z_i \mid \mathbf{x}, \mathbf{z}_{<i})}{p(z_i \mid \mathbf{x}, \mathbf{z}_{<i})} \right\}, \tag{2}$$

which guarantees that the resulting procedure samples exactly from the target distribution and thus preserves fidelity to the target model $q$. It accelerates generation by reducing the number of expensive target computations. The actual speedup is mainly determined by the acceptance rate and the cost gap between the draft and target model.

## 3 METHOD

The goal of SPEC-RL is to accelerate RL rollouts by avoiding redundant regeneration. Instead of sampling complete trajectories from scratch at every step, we leverage cached rollouts from the previous epoch and reuse as much of them as possible, only generating the minimal continuation that is inconsistent with the current policy (Figure 3). This reduces the number of decoded tokens and directly cuts rollout latency. The detailed procedure is described in Algorithm 1.

### 3.1 SPECULATIVE DECODING OVER CACHED ROLLOUTS WITH LENIENCE

At the core of SPEC-RL is adapting speculative decoding to the RL setting by treating cached rollouts as draft sequences. For a prompt $\mathbf{x}$, let $\mathbf{y}^{old} = \{y_i^{old}\}$ denote the cached rollout produced when this prompt was last seen in training. Instead of generating from scratch, we verify each cached token under the current policy and decide whether it can be reused. Formally, following the standard draft-and-verify formulation in Equation 2, we replace the draft distribution $p$ with the previous policy $\pi_{\text{old}}$ and the target distribution $q$ with the current policy $\pi_t$ at epoch $t$, yielding the acceptance rule

$$\alpha_i = \min\Big(1, \frac{\pi_t(y_i^{old} \mid \mathbf{x}, \mathbf{y}_{<i}^{old})}{\pi_{\text{old}}(y_i^{old} \mid \mathbf{x}, \mathbf{y}_{<i}^{old})}\Big). \tag{3}$$

While the vanilla rule ensures exact consistency with the current policy, it can be overly strict in practice, limiting the amount of reuse. To further improve reuse, we introduce a lenience parameter $\ell$ following prior work on speculative decoding (Chen et al., 2024). Lenience relaxes the acceptance condition, effectively shifting the decision boundary and permitting more tokens to be reused. Formally, the acceptance rule becomes

$$\tilde{\alpha}_i = \min\Big(1, \ell \cdot \frac{\pi_t(y_i^{old} \mid \mathbf{x}, \mathbf{y}_{<i}^{old})}{\pi_{\text{old}}(y_i^{old} \mid \mathbf{x}, \mathbf{y}_{<i}^{old})}\Big). \tag{4}$$

Each cached token is accepted if $u \sim \mathcal{U}(0, 1)$ satisfies $u \leq \tilde{\alpha}_i$, and rejected otherwise. When $\ell = 1$, this reduces to the vanilla speculative rule; $\ell > 1$ increases acceptance and yields longer reused prefixes; $\ell \to \infty$ corresponds to full reuse; and $\ell \to 0$ recovers standard RLVR without reuse. This simple knob provides a flexible way to balance rollout efficiency and exploration.

After applying the acceptance rule, the procedure identifies the first rejection position $n$. All tokens before this rejection position are retained as the verified prefix $\mathbf{y}_{<n}^{old}$, while the remaining suffix is discarded. The current policy $\pi_t$ then resumes generation from this point onward, producing a new suffix $\mathbf{y}_{\geq n}^{new}$. Finally, the verified prefix and the regenerated suffix are concatenated to form the new rollout $\mathbf{y}^{new}$. This end-to-end process—verification, generation, and assembly—is summarized in Algorithm 1.

---

**Algorithm 1:** SPEC-RL

**Input:** Current policy $\pi_t$; Prompt $\mathbf{x}$; old response $\mathbf{y}^{old} = \{y_i^{old}\}$ with probability $p^{old}$; lenience $\ell \geq 1$.

1 Compute probability *in parallel* $p_i^{new} \leftarrow \pi_t(y_i^{old} \mid \mathbf{x}, \mathbf{y}_{<i}^{old}), i = 1, ..., |\mathbf{y}^{old}|$;

2 Compute acceptance probability $\tilde{\alpha} = \min(1, \ell \cdot \frac{p^{new}}{p^{old}})$;

3 Initialize rejection position $n \leftarrow |\mathbf{y}^{old}| + 1$;

4 **for** $i = 1$ **to** $|\mathbf{y}^{old}|$ **do**

5      Sample $u \sim \mathcal{U}(0, 1)$;

6      **if** $u > \tilde{\alpha}_i$ **then**

7          Assign rejection position $n \leftarrow i$;

8          break;

9 Generate response $\mathbf{y}_{\geq n}^{new} \leftarrow \pi_t(\cdot \mid \mathbf{x}, \mathbf{y}_{<n}^{old})$;

10 Assemble response $\mathbf{y}^{new} \leftarrow \{\mathbf{y}_{<n}^{old}, \mathbf{y}_{\geq n}^{new}\}$;

11 **return** $\mathbf{y}^{new}$

---

## 3.2 IMPLEMENTING SPEC-RL IN RLVR TRAINING

To enable practical use in RLVR pipelines, SPEC-RL introduces a lightweight cache module that stores rollouts from the previous epoch and continuously refreshes them as training proceeds. When the same prompt reappears, its cached response is retrieved and verified under the current policy. Verified prefixes are reused directly, while rejected suffixes are scheduled for continuation. This reuse–continue mechanism is implemented in Algorithm 1, which shows how verified prefixes and regenerated suffixes are combined into the final response. For efficient batching, all requests are packed into a single call to the rollout engine. Verified prefixes and prompts are aligned through left padding, so that different requests can be processed in parallel without fragmentation. This design ensures that SPEC-RL operates as a drop-in module: it modifies only the rollout stage, requires no change to reward computation or policy updates, and is compatible with mainstream algorithms such as GRPO, PPO, and DAPO.

## 3.3 DISCUSSION

To further understand SPEC-RL, we discuss its connections and differences with both standard speculative decoding and existing RLVR training. This comparison helps situate the method more clearly and highlight its key contributions.

**Relation to speculative decoding.** SPEC-RL follows the draft–and–verify paradigm of speculative decoding, but in a simplified, single-round form. Vanilla speculative decoding typically requires a separate draft model, loading extra parameters, scheduling overhead, and multiple verification rounds. In contrast, SPEC-RL reuses the previous policy as the draft, with cached rollouts available "for free". The current policy $\pi_t$ performs only one parallel verification pass; after the first rejection, the suffix is generated directly. This eliminates the need for auxiliary models while preserving the fidelity guarantees of speculative decoding.

**Relation to vanilla RLVR.** Compared to standard RLVR, SPEC-RL modifies only the rollout stage. In vanilla training, every epoch regenerates full trajectories from scratch, even though large portions of tokens are already shared between consecutive epochs, as shown in Figure 2. SPEC-RL exploits this redundancy by verifying cached rollouts, reusing the accepted prefix, and regenerating only the suffix. Fully accepted responses can be reused without any generation, directly reducing rollout cost while ensuring consistency with the current policy.

**Why lenience matters.** RLVR training proceeds through incremental updates, so adjacent policies remain closely aligned. This makes lenience a natural fit: with moderate $\ell$ values, one can reuse tokens that are close to the current policy distribution without deviating significantly. Moreover, since the draft model corresponds to the model from the previous epoch, it remains naturally close to the current policy, thereby ensuring that cached rollouts are still informative. Such relaxation preserves learning signals while substantially reducing rollout cost, as parallel verification over cached rollouts is far cheaper than regenerating entire trajectories.

## 4 EXPERIMENTS

### 4.1 EXPERIMENT SETUP

We train our models using the verl (Sheng et al., 2025) framework with vLLM (Kwon et al., 2023) as the rollout engine, on data sampled from DeepMath (6,144 examples, denoted as DeepMath-6K) (He et al., 2025) and SimpleRL (8,192 examples, denoted as SimpleRL-8K) (Zeng et al., 2025). All experiments use a prompt batch size of 1,024 and a maximum response length of 4,096 tokens, conducted on a single node with $8\times$ NVIDIA H100 GPUs. Rollout is performed at a temperature of 1.0. The actor learning rate is fixed at $5 \times 10^{-7}$, and for PPO we set the critic learning rate to $1 \times 10^{-5}$.

**Benchmarks and metrics.** We evaluate rollout efficiency and accuracy on a broad suite of benchmarks. **Rollout efficiency** is reported as the number of generated tokens and the relative speedup (baseline time divided by method time). **Math reasoning benchmarks** include AMC 2023 (Art of

Problem Solving, 2024), GSM8K (Cobbe et al., 2021), MATH-500 (Hendrycks et al., 2021), Minerva Math (Lewkowycz et al., 2022a), and OlympiadBench (He et al., 2024). **Out-of-distribution (OOD) benchmarks** include MMLU-STEM (Hendrycks et al., 2020) and IFEval (Zhou et al., 2023), which evaluate the generalization capability of the model. Full hyperparameter and evaluation details are provided in Appendices A.1 and A.2.

## 4.2 MAIN PERFORMANCE

**Overall performance on various models and algorithms.** We evaluate SPEC-RL across multiple model families (Qwen, LLaMA) and RL algorithms (GRPO, PPO, DAPO), with results summarized in Table 1. Across nine model–algorithm settings, SPEC-RL yields an average speedup of $2.31\times$ by reducing generated tokens by 66The largest gain is with Qwen-3-8B-Base under DAPO $(1{,}052.2M \rightarrow 326.2M$ tokens; $2.88\times)$, while even the smallest case (Qwen-3-8B-Base with PPO) achieves $1.94\times$ without accuracy loss. These improvements closely track the reduction in generated tokens, confirming that token-level savings drive the acceleration.

On math benchmarks, accuracy remains broadly stable: larger models are highly robust, while smaller models show only small fluctuations. For OOD tasks, MMLU-STEM stays nearly unchanged, and IFEval improves in several cases—for example, +6.5 points on Qwen-3-8B-Base with GRPO. Overall, SPEC-RL accelerates rollout generation substantially without degrading reasoning quality, and sometimes even improves out-of-distribution generalization.

Table 1: Overall results across models (Qwen, LLaMA) and algorithms (GRPO, PPO, DAPO) on DeepMath-6K. For each model size and family, we report the performance of its base model, the results of different RL algorithms, and the corresponding rollout efficiency and accuracy when equipped with SPEC-RL.

| Algorithm | Rollout Efficiency | | Math Reasoning | | | | | OOD | | AVG |
|---|---|---|---|---|---|---|---|---|---|---|
| | Tokens (M) | Speedup | AMC23 | GSM8K | MATH 500 | Minerva Math | Olympiad Bench | MMLU STEM | IFEval | |
| *Qwen-3-1.7B-Base* | | | | | | | | | | |
| Base Model | - | - | 22.5 | 59.1 | 45.0 | 12.5 | 16.7 | 39.3 | 17.9 | 30.4 |
| GRPO | 554.8 | 1.00× | **42.5** | 82.6 | 64.4 | 26.5 | 25.5 | **60.7** | 24.4 | 46.7 |
| ↪ + SPEC-RL | **182.7** | **2.29×** | 37.5 | **84.4** | **68.0** | **29.4** | **29.3** | 58.3 | **28.8** | **48.0** |
| PPO | 565.1 | 1.00× | **35.0** | 82.0 | 63.0 | 26.8 | 25.3 | **59.4** | 25.5 | 45.3 |
| ↪ + SPEC-RL | **230.8** | **1.94×** | 35.0 | 82.0 | 64.8 | 25.4 | 25.9 | 58.6 | 25.9 | 45.4 |
| DAPO | 543.1 | 1.00× | **30.0** | 79.6 | **60.8** | 24.6 | 23.0 | 52.2 | 24.8 | **42.1** |
| ↪ + SPEC-RL | **171.6** | **2.17×** | 22.5 | **80.1** | 60.0 | **25.7** | **25.5** | **53.5** | **27.0** | 42.0 |
| *Qwen-3-8B-Base* | | | | | | | | | | |
| Base Model | - | - | 40.0 | 83.0 | 67.4 | 27.2 | 34.1 | 60.4 | 29.9 | 48.9 |
| GRPO | 1033.1 | 1.00× | **75.0** | 94.1 | 86.4 | 43.8 | **53.0** | **84.6** | 41.2 | 68.3 |
| ↪ + SPEC-RL | **336.6** | **2.51×** | 70.0 | **94.5** | **87.8** | **44.1** | 51.0 | 84.5 | **47.7** | **68.5** |
| PPO | 984.0 | 1.00× | 70.0 | **94.2** | **85.8** | 43.0 | **51.6** | 83.8 | 41.6 | 67.1 |
| ↪ + SPEC-RL | **400.1** | **1.94×** | **75.0** | 92.9 | 85.2 | **43.4** | 50.8 | **84.4** | 41.0 | **67.5** |
| DAPO | 1052.2 | 1.00× | **75.0** | 93.3 | **84.8** | 40.1 | 48.6 | **82.4** | 39.6 | 66.3 |
| ↪ + SPEC-RL | **326.2** | **2.88×** | 65.0 | **93.8** | 84.4 | **43.8** | **50.4** | 82.2 | **44.4** | 66.3 |
| *LLaMA-3.2-1B-Instruct* | | | | | | | | | | |
| Base Model | - | - | 0.0 | 26.7 | 14.2 | 4.0 | 2.8 | 32.6 | 37.0 | 16.8 |
| GRPO | 553.9 | 1.00× | 5.0 | 28.1 | 19.2 | **3.3** | 4.9 | 33.1 | 37.0 | 18.7 |
| ↪ + SPEC-RL | **162.5** | **2.60×** | 7.5 | **28.7** | **19.4** | 1.8 | **5.0** | **34.5** | **37.2** | **19.2** |
| PPO | 521.5 | 1.00× | 10.0 | 31.6 | **20.8** | 4.0 | **6.4** | 34.3 | **42.7** | **21.4** |
| ↪ + SPEC-RL | **210.6** | **2.01×** | 10.0 | **32.4** | 20.2 | **5.5** | 5.0 | **35.3** | 40.7 | 21.3 |
| DAPO | 482.6 | 1.00× | 7.5 | 29.6 | 19.2 | **4.0** | **5.5** | 33.0 | **38.6** | 19.6 |
| ↪ + SPEC-RL | **123.1** | **2.48×** | 10.0 | **34.9** | 20.2 | 4.0 | **5.5** | **35.5** | 38.4 | **21.2** |

Table 2: Comparison between SPEC-RL and a random reuse baseline on GRPO. In the random reuse setting, the rejection position for each sequence is drawn uniformly at random, resulting in roughly half of the tokens being reused on average.

| Algorithm | Rollout Efficiency | | Math Reasoning | | | | | OOD | | AVG |
|---|---|---|---|---|---|---|---|---|---|---|
| | Tokens (M) | Speedup | AMC23 | GSM8K | MATH 500 | Minerva Math | Olympiad Bench | MMLU STEM | IFEval | |
| GRPO | 554.8 | 1.00× | **42.5** | 82.6 | 64.4 | 26.5 | 25.5 | **60.7** | 24.4 | 46.7 |
| ↪ + Random Reuse | 304.5 | **2.35×** | 37.5 | 80.0 | 60.4 | 21.7 | 25.3 | 53.1 | 24.0 | 43.1 |
| ↪ + SPEC-RL | **182.7** | 2.29× | 37.5 | **84.4** | **68.0** | **29.4** | **29.3** | 58.3 | **28.8** | **48.0** |

Table 3: Ablation on lenience parameter $\ell$ on the DeepMath-6K. Here $\ell = 1$ corresponds to vanilla speculative decoding, while $\ell = \infty$ corresponds to full reuse.

| Algorithm | Rollout Efficiency | | Math Reasoning | | | | | OOD | | AVG |
| | Tokens (M) | Speedup | AMC23 | GSM8K | MATH 500 | Minerva Math | Olympiad Bench | MMLU STEM | IFEval | |
|---|---|---|---|---|---|---|---|---|---|---|
| GRPO | 554.8 | 1.00× | **42.5** | 82.6 | 64.4 | 26.5 | 25.5 | 60.7 | 24.4 | 46.7 |
| ↪ + SPEC-RL $\ell = 1$ | 419.1 | 1.22× | 40.0 | 81.8 | 63.8 | 28.7 | 26.5 | 59.6 | 25.9 | 46.6 |
| ↪ + SPEC-RL $\ell = e^{0.2}$ | 246.7 | 1.86× | 37.5 | 83.3 | 66.4 | **29.8** | **29.6** | 58.5 | 25.9 | 47.3 |
| ↪ + SPEC-RL $\ell = e^{0.5}$ | 182.7 | 2.29× | 37.5 | **84.4** | **68.0** | 29.4 | 29.3 | 58.3 | 28.8 | **48.0** |
| ↪ + SPEC-RL $\ell = e^{0.8}$ | 144.8 | 2.64× | 37.5 | 83.5 | 63.6 | 27.2 | 25.0 | **61.7** | 26.2 | 46.4 |
| ↪ + SPEC-RL $\ell = e^{1.0}$ | 123.0 | 2.91× | 37.5 | 83.9 | 62.4 | 25.7 | 24.9 | 54.8 | 28.3 | 45.4 |
| ↪ + SPEC-RL $\ell = e^{2.0}$ | 114.4 | 3.05× | 30.0 | 80.4 | 55.0 | 21.0 | 21.9 | 53.5 | **29.0** | 41.5 |
| ↪ + SPEC-RL $\ell = \infty$ | **40.0** | **14.86×** | 32.5 | 78.1 | 60.4 | 19.9 | 23.7 | 44.1 | 22.0 | 40.1 |

Additional analyses—including per-step training curves, wall-clock breakdowns, comparisons on DeepMath-6K and SimpleRL-8K, and the effect of training-set size—are provided in Appendices A.3, A.4, A.5, and A.6.

**Comparison with random reuse strategy.** We further compare SPEC-RL with a random reuse baseline, where rejection positions are sampled uniformly at random, leading to roughly half of the tokens being reused on average. As shown in Table 2, random reuse reduces rollout cost (304.5M vs. 554.8M tokens) and improves efficiency (2.35× speedup), but causes a substantial drop in accuracy (43.1 vs. 46.7). In particular, it degrades performance on high-stakes benchmarks such as MATH-500 (60.4 vs. 64.4) and Minerva Math (21.7 vs. 26.5). By contrast, SPEC-RL achieves comparable or better efficiency gains (182.7M tokens, 2.29× speedup) while preserving accuracy. This contrast highlights that naive reuse introduces harmful noise, whereas SPEC-RL leverages speculative verification to retain policy fidelity while accelerating training. The detailed intermediate training results of random reuse are reported in Appendix A.7

## 4.3 ABLATION STUDY

We conduct ablation experiments on Qwen-3-1.7B-Base with GRPO using the DeepMath-6K dataset with a batch size of 1,024. Under this setting, one epoch corresponds to 6 steps, and the results are summarized in Table 3 and Figures 4, 5, and 6.

**Impact of lenience $\ell$.** As shown in Table 3, increasing $\ell$ consistently improves rollout efficiency: starting from vanilla speculative decoding at $\ell = 1$ with a speedup of only 1.22×, the acceleration rises steadily and reaches 14.86× when $\ell \to \infty$. Accuracy, however, does not follow the same

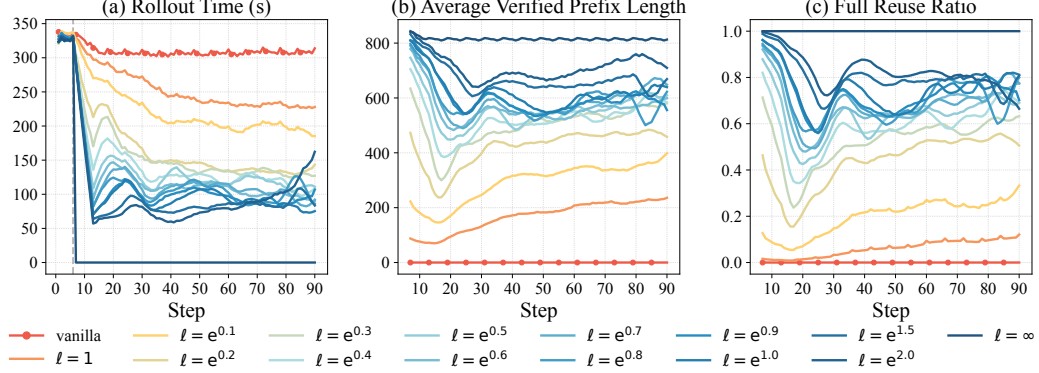

Figure 4: Training dynamics of SPEC-RL under different $\ell$. (a) Rollout time per training step decreases as $\ell$ increases, where the dashed line indicates the step at which speculative decoding begins. (b) Average verified prefix length grows both with larger $\ell$ and across training steps, reflecting stronger policy alignment. (c) Full reuse ratio—the fraction of samples fully reusing cached rollouts—also rises, complementing prefix length and jointly explaining the observed efficiency gains.

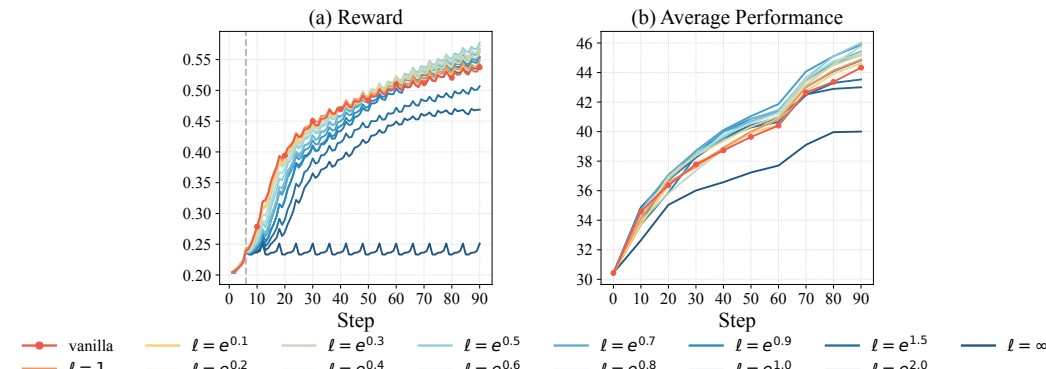

Figure 5: Effect of lenience $\ell$ on learning outcomes. (a) Reward trajectories show that moderate $\ell$ maintains stability and eventually surpasses vanilla GRPO, while overly large $\ell$ slows progress. (b) Average performance follows the same trend: moderate $\ell$ preserves accuracy, whereas aggressive reuse ($\ell \geq e^{1.0}$) degrades both reward and final performance.

trend—performance peaks at $\ell = e^{0.5}$ with 48.0, but declines when reuse becomes overly aggressive, dropping to 40.1 at $\ell \to \infty$. Overall, moderate lenience values strike the best balance, yielding 2–3× rollout speedups while preserving or slightly improving accuracy, whereas extreme reuse sacrifices performance despite dramatic acceleration. The detailed intermediate results throughout training are provided in Appendix A.8.

**Acceleration is jointly driven by lenience and policy alignment.** Cached responses become reusable from the second epoch (e.g., step 7), at which point rollout time shows the first sharp drop (Figure 4 (a)). Before that, all curves overlap because no old policy exists. Once reuse starts, the SPEC-RL curves immediately diverge and rollout time falls substantially. Along the lenience axis, vanilla GRPO stabilizes near 300s per step, while even the default setting ($\ell = 1$) lowers this to slightly above 200s. Larger $\ell$ yields further gains—for example, at $\ell = e^{0.2}$ the later rollout time is already below half of the baseline ( 150s), with even more reduction for higher $\ell$ (Figure 4 (a)). Correspondingly, both accepted-prefix lengths and skip ratios grow with increasing $\ell$ (Figure 4 (b,c)). Along the training axis, accepted-prefix length is initially high, dips around steps 10–20 due to early policy shifts, and then rises again as the policy aligns (Figure 4 (b)). Meanwhile, the skip ratio increases steadily throughout training (Figure 4 (c)). Together, these patterns show that efficiency gains in SPEC-RL stem from both lenience-controlled acceptance and the growing alignment between successive policies.

**Excessive reuse stresses optimization.** When $\ell \to \infty$, reuse becomes complete and every cached response is fully reused from the second epoch onward. As shown in Figure 6 (a–c), entropy, KL loss, and the gradient clipping ratio all rise dramatically compared with vanilla GRPO and settings with $\ell \leq e^{2.0}$, quickly shooting beyond the plotting range, indicating severe instability. Because exploration collapses under complete reuse, the outputs of all subsequent epochs become identical, and the training reward exhibits a clear cyclic fluctuation with the period of one epoch (6 steps in our setup), as illustrated in Figure 5 (a). These unstable dynamics further translate into a sharp degradation of downstream accuracy: the average math performance drops markedly (Figure 5 (b) and Table 3). Overall, extreme acceleration from complete reuse comes at the cost of exploration collapse and unstable optimization dynamics.

**Moderate reuse preserves healthy learning signals.** In contrast to the instability observed at extreme reuse, moderate lenience values (around $\ell = e^{0.5}$) maintain well-behaved optimization dynamics. As shown in Figure 6 (a–c), entropy and KL loss remain close to those of vanilla GRPO, and clipping is not abnormally triggered. Meanwhile, reward trajectories under moderate $\ell$ maintain stability and eventually surpass the baseline (Figure 5 (a)), while average math performance is preserved or slightly improved (Figure 5 (b)), consistent with the peak average score of 48.0 reported in Table 3. These results demonstrate that moderate lenience values enable 2–3× acceleration without

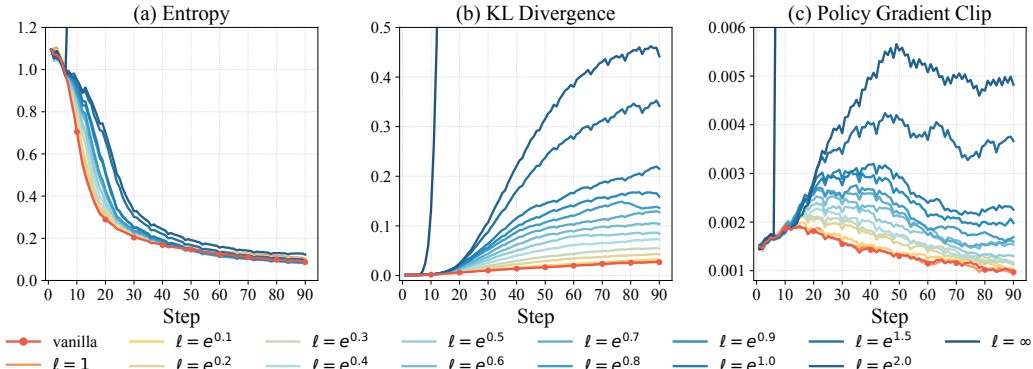

Figure 6: Training dynamics of SPEC-RL under different $\ell$. (a) Entropy decreases more rapidly with larger $\ell$. (b) KL Divergence gradually rises with more steps, especially under larger $\ell$. (c) Policy Gradient Clip ratio first increases and then stabilizes, with higher $\ell$ leading to larger values.

sacrificing reward signals or downstream reasoning accuracy. Case studies in Appendix B further show how SPEC-RL reuses verified prefixes while keeping the reasoning chain intact, illustrating why moderate reuse achieves the best trade-off.

## 5 RELATED WORK

**Efficiency in RLVR.** Prior work improves RLVR efficiency mostly at the *trajectory level*—via parallel rollouts (Xu et al., 2025), modified objectives (Brantley et al., 2025; Lin et al., 2025), data-centric batch restructuring (Liu et al., 2025; Zhang et al., 2025b), system heuristics (Yu et al., 2025; Zheng et al., 2025), or trajectory replay (Zhang et al., 2025a). These approaches reduce the number of trajectories or stabilize optimization, but do not shrink the *token count per trajectory*, which dominates rollout time. Our method targets this missing dimension by reusing verified prefixes, improving token-level efficiency while keeping RLVR objectives unchanged.

**Speculative decoding.** Speculative decoding proposes tokens with a draft model and verifies them with a stronger model (Leviathan et al., 2023), with later variants improving parallelism or multi-token acceptance (Cai et al., 2024; Chen et al., 2024; Qin et al., 2024; Sun et al., 2024). While designed for inference and typically requiring extra modules, we repurpose speculative decoding for RLVR by using the previous policy's outputs as drafts, enabling prefix reuse without additional models.

## 6 CONCLUSION

We address the rollout bottleneck in reinforcement learning with verifiable rewards (RLVR) by introducing SPEC-RL, which integrates speculative decoding into rollout generation. Instead of regenerating trajectories from scratch, SPEC-RL treats previous-epoch rollouts as implicit drafts: tokens are verified under the current policy to form a verified prefix, then generation resumes from the first rejection position. A lenience parameter $\ell$ modulates the acceptance rule, trading off reuse and exploration. Experiments on Qwen and LLaMA models with GRPO, PPO, and DAPO show consistent $2$–$3\times$ rollout speedups with largely preserved, and sometimes improved, reasoning and OOD performance. This demonstrates that rollout redundancy can be systematically exploited without modifying objectives, rewards, or update rules. Limitations include dependence on cached responses and potential under-exploration with high lenience. Future work includes adaptive scheduling of $\ell$ and extending speculative reuse to multi-turn and large-scale RLVR settings. Overall, SPEC-RL offers a simple, model-agnostic way to substantially reduce the cost of RLVR training.

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

# A    MORE DETAILS OF SPEC-RL

This appendix provides additional details on experimental settings, hyperparameters, and reward design, as well as extended ablation studies and full step-level results. We begin with shared training configurations and evaluation setups, then report intermediate training trajectories, efficiency analyses, and end-to-end time breakdowns. Finally, we present ablations across datasets (DeepMath-6K vs. SimpleRL-8K) and training-set sizes (2K–6K), additional baseline comparisons, and case studies that illustrate the behavior of SPEC-RL in practice.

## A.1    HYPERPARAMETERS

We report the shared training settings (model families, rollout engine, batch size, sequence lengths, training steps, and optimizer details), as well as the algorithm-specific configurations. All experiments use Qwen-3-1.7B-Base, Qwen-3-8B-Base, and LLaMA-3.2-1B as backbone models. Rollouts are generated using vLLM (rollout $N = 8$) with a global batch size of 1024. The maximum prompt length is 1,024 tokens, and the maximum response length is 4,096 tokens. For optimization, the actor is trained using AdamW (learning rate $5 \times 10^{-7}$, weight decay 0.01, and gradient clipping of 1.0). For PPO, the critic is additionally optimized with AdamW (learning rate $1 \times 10^{-5}$, weight decay 0.01, clipping 1.0). Algorithm-specific differences are as follows. GRPO enables KL regularization with a coefficient of 0.0001, whereas PPO and DAPO disable KL regularization. DAPO further adopts a wider clipping range (high = 0.28, $c = 10$) compared to GRPO and PPO (high = 0.2, $c = 3$). Additionally, DAPO utilizes dynamic sampling. To ensure fair comparison with GRPO and PPO, we control for the total amount of rollout data: each training step in DAPO corresponds to multiple generation steps, and the evaluation interval is reduced from every 10 steps to every 5 steps. SPEC-RL uses default lenience values of $e^{0.5}$ for GRPO, $e^{0.3}$ for PPO, and $e^{0.15}$ for DAPO, chosen via grid search to balance rollout efficiency and stability. All methods employ the `math-verify` reward, which assigns $+1$ if the final boxed or numeric answer matches the ground truth and 0 otherwise. This simple, deterministic design ensures that the reward is aligned with evaluation metrics across benchmarks.

We use a rule-based reward function that depends solely on the correctness of the final answer. Specifically, we utilize the `math-verify` library to verify each generated solution: if the predicted answer matches the reference, the model receives a reward of $+1$, and otherwise, 0. The `math-verify` library is responsible for parsing the model output, extracting the final boxed or numeric answer, and checking it against the ground truth. No format-based shaping or auxiliary heuristics are used. This choice maintains a simple, deterministic, and aligned reward signal across all benchmarks, aligning with the evaluation objective.

## A.2    DETAILED EVALUATION SETUPS

Our evaluation setup largely follows prior work (Zeng et al., 2025; Yang et al., 2024), ensuring consistency and comparability with established baselines. For all math reasoning benchmarks, including AMC23, GSM8K, MATH-500, Minerva Math, OlympiadBench, and MMLU-STEM, we use a maximum generation length of 16,000 tokens, with nucleus sampling ($p = 0.95$) and temperature set to 1.0. For IFEval, we employ the lighteval (Habib et al., 2023) framework for evaluation, maintaining the same decoding parameters as those used in the math reasoning benchmarks. This uniform setup ensures that all comparisons focus on the effects of SPEC-RL, rather than variations in decoding configurations. For experiments on DeepMath-6K, we report the performance at step 90 (corresponding to 15 epochs with 6,144 examples and a batch size of 1,024). For SimpleRL-8K, we report the performance at step 100.

## A.3    PERFORMANCE OVER TRAINING STEPS

To provide a more complete view of model behavior and enhance the robustness of our method, we also report performance trajectories throughout training. For each setting, results are shown every 10 steps, comparing the vanilla algorithm with its SPEC-RL variant, as shown in Tables 4, 5, 6, 7, 8, 9, 10, 11, 12. This step-wise view complements the main results by illustrating how rollout efficiency and accuracy evolve consistently during training, rather than only at the final checkpoint.

Table 4: Intermediate training results of Qwen-3-1.7B-Base on DeepMath-6K with GRPO and SPEC-RL. We report rollout efficiency and accuracy every 10 training steps, with GRPO and its SPEC-RL variant interleaved.

| Algorithm | Step | Rollout Efficiency | | Math Reasoning | | | | | OOD | | AVG |
|---|---|---|---|---|---|---|---|---|---|---|---|
| | | Tokens (M) | Speedup | AMC23 | GSM8K | MATH 500 | Minerva Math | Olympiad Bench | MMLU STEM | IFEval | |
| Base Model | 0 | - | - | 22.5 | 59.1 | 45.0 | 12.5 | 16.7 | 39.3 | 17.9 | 30.4 |
| GRPO | 10 | 65.8 | 1.00× | 37.5 | 74.0 | 54.0 | 18.8 | 22.8 | 43.9 | 20.3 | 38.8 |
| ↪ + SPEC-RL | 10 | 43.6 | 1.41× | 27.5 | 75.7 | 55.8 | 21.7 | 21.8 | 43.0 | 22.4 | 38.3 |
| GRPO | 20 | 127.4 | 1.00× | 27.5 | 78.2 | 57.8 | 25.4 | 24.7 | 45.5 | 20.3 | 39.9 |
| ↪ + SPEC-RL | 20 | 67.2 | 1.66× | 30.0 | 80.0 | 63.0 | 25.0 | 24.6 | 46.9 | 22.2 | 41.7 |
| GRPO | 30 | 187.7 | 1.00× | 37.5 | 80.1 | 60.4 | 22.4 | 25.0 | 47.9 | 20.9 | 42.0 |
| ↪ + SPEC-RL | 30 | 85.1 | 1.85× | 30.0 | 81.0 | 64.0 | 25.4 | 27.6 | 51.1 | 26.2 | 43.6 |
| GRPO | 40 | 248.3 | 1.00× | 32.5 | 79.9 | 65.0 | 23.5 | 24.7 | 50.7 | 21.3 | 42.5 |
| ↪ + SPEC-RL | 40 | 102.9 | 1.96× | 37.5 | 80.7 | 63.8 | 26.5 | 26.1 | 52.2 | 23.5 | 44.3 |
| GRPO | 50 | 309.1 | 1.00× | 35.0 | 81.2 | 64.2 | 25.7 | 24.6 | 53.7 | 25.1 | 44.2 |
| ↪ + SPEC-RL | 50 | 119.4 | 2.06× | 32.5 | 81.1 | 64.4 | 28.7 | 28.0 | 55.6 | 27.7 | 45.4 |
| GRPO | 60 | 370.4 | 1.00× | 35.0 | 81.3 | 63.6 | 28.3 | 26.7 | 56.0 | 24.0 | 45.0 |
| ↪ + SPEC-RL | 60 | 135.1 | 2.14× | 27.5 | 83.4 | 66.4 | 26.5 | 29.9 | 54.7 | 28.8 | 45.3 |
| GRPO | 70 | 431.9 | 1.00× | 42.5 | 82.3 | 61.8 | 26.5 | 28.1 | 55.5 | 26.2 | 46.1 |
| ↪ + SPEC-RL | 70 | 153.2 | 2.18× | 40.0 | 82.5 | 65.4 | 26.5 | 29.9 | 55.6 | 27.4 | 46.8 |
| GRPO | 80 | 493.5 | 1.00× | 25.0 | 82.0 | 64.4 | 24.3 | 26.4 | 59.4 | 25.0 | 43.8 |
| ↪ + SPEC-RL | 80 | 168.1 | 2.24× | 45.0 | 83.7 | 67.0 | 29.8 | 29.6 | 57.1 | 28.3 | 48.6 |
| GRPO | 90 | 554.8 | 1.00× | 42.5 | 82.6 | 64.4 | 26.5 | 25.5 | 60.7 | 24.4 | 46.7 |
| ↪ + SPEC-RL | 90 | 182.7 | 2.29× | 37.5 | 84.4 | 68.0 | 29.4 | 29.3 | 58.3 | 28.8 | 48.0 |

Table 5: Intermediate training results of Qwen-3-8B-Base on DeepMath-6K with GRPO and SPEC-RL. We report rollout efficiency and accuracy every 10 training steps, with GRPO and its SPEC-RL variant interleaved.

| Algorithm | Step | Rollout Efficiency | | Math Reasoning | | | | | OOD | | AVG |
|---|---|---|---|---|---|---|---|---|---|---|---|
| | | Tokens (M) | Speedup | AMC23 | GSM8K | MATH 500 | Minerva Math | Olympiad Bench | MMLU STEM | IFEval | |
| Base Model | 0 | - | - | 40.0 | 83.0 | 67.4 | 27.2 | 34.1 | 60.4 | 29.9 | 48.9 |
| GRPO | 10 | 71.9 | 1.00× | 60.0 | 91.5 | 80.0 | 32.7 | 44.3 | 64.7 | 34.4 | 58.2 |
| ↪ + SPEC-RL | 10 | 53.0 | 1.36× | 60.0 | 92.0 | 80.0 | 37.1 | 43.9 | 64.2 | 37.2 | 59.2 |
| GRPO | 20 | 158.2 | 1.00× | 62.5 | 93.6 | 82.8 | 40.4 | 49.3 | 77.3 | 39.0 | 63.6 |
| ↪ + SPEC-RL | 20 | 76.4 | 1.96× | 65.0 | 93.3 | 83.6 | 42.6 | 48.6 | 72.2 | 43.4 | 64.1 |
| GRPO | 30 | 278.1 | 1.00× | 70.0 | 92.7 | 84.2 | 39.7 | 48.9 | 80.4 | 35.7 | 64.5 |
| ↪ + SPEC-RL | 30 | 116.7 | 2.18× | 65.0 | 93.5 | 85.0 | 43.0 | 49.5 | 80.4 | 47.9 | 66.3 |
| GRPO | 40 | 404.2 | 1.00× | 67.5 | 93.5 | 85.2 | 40.8 | 50.2 | 82.0 | 37.9 | 65.3 |
| ↪ + SPEC-RL | 40 | 156.4 | 2.31× | 75.0 | 94.1 | 84.2 | 44.5 | 49.0 | 83.3 | 46.8 | 68.1 |
| GRPO | 50 | 532.0 | 1.00× | 70.0 | 93.5 | 85.4 | 42.6 | 49.5 | 82.8 | 40.1 | 66.3 |
| ↪ + SPEC-RL | 50 | 194.4 | 2.36× | 77.5 | 93.3 | 84.8 | 44.1 | 52.3 | 83.2 | 45.7 | 68.7 |
| GRPO | 60 | 659.3 | 1.00× | 72.5 | 93.1 | 84.8 | 44.1 | 51.4 | 83.0 | 38.8 | 66.8 |
| ↪ + SPEC-RL | 60 | 235.7 | 2.36× | 72.5 | 94.4 | 85.4 | 43.0 | 51.1 | 84.4 | 44.9 | 68.0 |
| GRPO | 70 | 785.6 | 1.00× | 65.0 | 93.3 | 84.8 | 43.4 | 51.3 | 84.3 | 34.8 | 65.3 |
| ↪ + SPEC-RL | 70 | 279.2 | 2.36× | 62.5 | 94.4 | 87.0 | 43.8 | 51.7 | 84.7 | 47.5 | 67.4 |
| GRPO | 80 | 910.2 | 1.00× | 67.5 | 94.0 | 85.8 | 43.4 | 50.2 | 84.7 | 40.1 | 66.5 |
| ↪ + SPEC-RL | 80 | 311.1 | 2.42× | 75.0 | 93.4 | 87.4 | 43.4 | 52.1 | 85.2 | 48.2 | 69.2 |
| GRPO | 90 | 1033.1 | 1.00× | 75.0 | 94.1 | 86.4 | 43.8 | 53.0 | 84.6 | 41.2 | 68.3 |
| ↪ + SPEC-RL | 90 | 336.6 | 2.51× | 70.0 | 94.5 | 87.8 | 44.1 | 51.0 | 84.5 | 47.7 | 68.5 |

**Training Dynamics and Efficiency Across Different RL Algorithms.** We present the efficiency of our method across RL algorithms in Figures 7, 8, and compare rewards and rollout time against baselines in Figures 9, 10. Across all three algorithms, SPEC-RL substantially reduces rollout time while preserving learning quality: rewards match or exceed the vanilla baselines under PPO and GRPO, and are largely on par under DAPO (with a minor late-stage gap on Qwen3-8B). The efficiency gains align with stronger speculative reuse signals: the full reuse ratio quickly rises and stabilizes around 0.6–0.85 after early transients, and the average verified prefix length remains large (hundreds to 1.2k tokens) and generally increases over training—most prominently on Qwen3-8B for GRPO/DAPO. Together, these curves indicate that SPEC-RL learns to reuse long, verified prefixes, trading decoding for reuse, which yields lower per-step generation cost without compromising reward progress.

Table 6: Intermediate training results of LLaMA-3.2-1B-Instruct on DeepMath-6K with GRPO and SPEC-RL. We report rollout efficiency and accuracy every 10 training steps, with GRPO and its SPEC-RL variant interleaved.

| Algorithm | Step | Rollout Efficiency | | Math Reasoning | | | | | OOD | | AVG |
| | | Tokens (M) | Speedup | AMC23 | GSM8K | MATH 500 | Minerva Math | Olympiad Bench | MMLU STEM | IFEval | |
|---|---|---|---|---|---|---|---|---|---|---|---|
| Base Model | 0 | - | - | 0.0 | 26.7 | 14.2 | 4.0 | 2.8 | 32.6 | 37.0 | 16.8 |
| GRPO | 10 | 72.1 | 1.00× | 7.5 | 27.2 | 14.2 | 2.9 | 3.4 | 31.7 | 38.8 | 18.0 |
| ↪ + SPEC-RL | 10 | 47.8 | 1.38× | 5.0 | 27.1 | 12.8 | 2.6 | 3.4 | 32.8 | 38.6 | 17.5 |
| GRPO | 20 | 141.2 | 1.00× | 5.0 | 28.3 | 15.6 | 2.6 | 3.9 | 33.5 | 39.7 | 18.4 |
| ↪ + SPEC-RL | 20 | 78.5 | 1.57× | 7.5 | 28.7 | 18.0 | 3.7 | 3.9 | 35.0 | 38.6 | 19.3 |
| GRPO | 30 | 204.5 | 1.00× | 5.0 | 27.1 | 17.4 | 2.9 | 4.4 | 35.1 | 38.3 | 18.6 |
| ↪ + SPEC-RL | 30 | 100.7 | 1.73× | 5.0 | 32.3 | 18.4 | 2.6 | 4.9 | 33.4 | 40.5 | 19.6 |
| GRPO | 40 | 266.8 | 1.00× | 10.0 | 29.5 | 15.4 | 3.3 | 4.4 | 33.2 | 41.2 | 19.6 |
| ↪ + SPEC-RL | 40 | 115.0 | 1.94× | 10.0 | 31.2 | 18.6 | 4.4 | 5.5 | 34.2 | 38.8 | 20.4 |
| GRPO | 50 | 326.2 | 1.00× | 12.5 | 27.9 | 17.6 | 3.7 | 5.3 | 34.3 | 38.1 | 19.9 |
| ↪ + SPEC-RL | 50 | 126.3 | 2.12× | 7.5 | 31.5 | 20.2 | 4.4 | 4.7 | 36.0 | 39.6 | 20.6 |
| GRPO | 60 | 382.9 | 1.00× | 15.0 | 28.4 | 17.8 | 3.3 | 5.2 | 34.0 | 40.5 | 20.6 |
| ↪ + SPEC-RL | 60 | 134.7 | 2.29× | 7.5 | 31.8 | 19.0 | 4.4 | 5.5 | 35.6 | 37.9 | 20.2 |
| GRPO | 70 | 438.3 | 1.00× | 15.0 | 30.1 | 17.6 | 4.8 | 5.5 | 34.6 | 37.3 | 20.7 |
| ↪ + SPEC-RL | 70 | 143.6 | 2.41× | 12.5 | 29.7 | 19.8 | 5.1 | 5.8 | 36.1 | 37.2 | 20.9 |
| GRPO | 80 | 495.3 | 1.00× | 5.0 | 25.9 | 17.6 | 4.0 | 3.9 | 33.3 | 37.2 | 18.1 |
| ↪ + SPEC-RL | 80 | 152.8 | 2.52× | 7.5 | 29.3 | 19.4 | 2.9 | 3.9 | 35.6 | 42.7 | 20.2 |
| GRPO | 90 | 553.9 | 1.00× | 5.0 | 28.1 | 19.2 | 3.3 | 4.9 | 33.1 | 37.0 | 18.7 |
| ↪ + SPEC-RL | 90 | 162.5 | 2.60× | 7.5 | 28.7 | 19.4 | 1.8 | 5.0 | 34.5 | 37.2 | 19.2 |

Table 7: Intermediate training results of Qwen-3-1.7B-Base on DeepMath-6K with PPO and SPEC-RL. We report rollout efficiency and accuracy every 10 training steps, with PPO and its SPEC-RL variant interleaved.

| Algorithm | Step | Rollout Efficiency | | Math Reasoning | | | | | OOD | | AVG |
| | | Tokens (M) | Speedup | AMC23 | GSM8K | MATH 500 | Minerva Math | Olympiad Bench | MMLU STEM | IFEval | |
|---|---|---|---|---|---|---|---|---|---|---|---|
| Base Model | 0 | - | - | 22.5 | 59.1 | 45.0 | 12.5 | 16.7 | 39.3 | 17.9 | 30.4 |
| PPO | 10 | 66.6 | 1.00× | 35.0 | 71.3 | 54.8 | 19.9 | 21.3 | 42.6 | 19.2 | 37.7 |
| ↪ + SPEC-RL | 10 | 46.5 | 1.34× | 27.5 | 73.2 | 56.8 | 17.6 | 23.3 | 42.9 | 19.4 | 37.2 |
| PPO | 20 | 129.2 | 1.00× | 35.0 | 77.3 | 60.4 | 22.8 | 25.0 | 46.6 | 21.8 | 41.3 |
| ↪ + SPEC-RL | 20 | 80.0 | 1.44× | 37.5 | 78.8 | 58.0 | 23.2 | 23.4 | 46.9 | 20.9 | 41.2 |
| PPO | 30 | 191.5 | 1.00× | 37.5 | 78.5 | 59.4 | 22.8 | 26.5 | 47.8 | 19.6 | 41.7 |
| ↪ + SPEC-RL | 30 | 106.2 | 1.56× | 37.5 | 78.3 | 62.6 | 23.5 | 25.6 | 49.8 | 22.7 | 42.9 |
| PPO | 40 | 253.6 | 1.00× | 40.0 | 77.7 | 61.4 | 23.5 | 25.3 | 50.7 | 22.7 | 43.0 |
| ↪ + SPEC-RL | 40 | 126.2 | 1.69× | 37.5 | 80.4 | 63.2 | 22.4 | 27.3 | 51.3 | 23.7 | 43.7 |
| PPO | 50 | 315.7 | 1.00× | 35.0 | 79.5 | 61.8 | 26.8 | 25.6 | 51.5 | 21.8 | 43.1 |
| ↪ + SPEC-RL | 50 | 157.3 | 1.68× | 40.0 | 80.9 | 64.4 | 26.1 | 25.9 | 54.2 | 26.6 | 45.4 |
| PPO | 60 | 377.7 | 1.00× | 27.5 | 81.6 | 63.8 | 29.4 | 26.8 | 53.9 | 23.3 | 43.8 |
| ↪ + SPEC-RL | 60 | 172.0 | 1.79× | 35.0 | 82.0 | 64.2 | 23.5 | 27.0 | 53.8 | 24.4 | 44.3 |
| PPO | 70 | 440.0 | 1.00× | 35.0 | 79.5 | 60.6 | 25.7 | 26.7 | 55.0 | 22.9 | 43.6 |
| ↪ + SPEC-RL | 70 | 194.7 | 1.83× | 35.0 | 80.7 | 65.8 | 27.6 | 26.8 | 55.1 | 25.9 | 45.3 |
| PPO | 80 | 503.0 | 1.00× | 45.0 | 81.4 | 63.8 | 25.4 | 29.3 | 58.6 | 23.7 | 46.7 |
| ↪ + SPEC-RL | 80 | 207.2 | 1.93× | 40.0 | 82.6 | 63.6 | 29.8 | 28.0 | 54.3 | 25.7 | 46.3 |
| PPO | 90 | 565.1 | 1.00× | 35.0 | 82.0 | 63.0 | 26.8 | 25.3 | 59.4 | 25.5 | 45.3 |
| ↪ + SPEC-RL | 90 | 230.8 | 1.94× | 35.0 | 82.0 | 64.8 | 25.4 | 25.9 | 58.6 | 25.9 | 45.4 |

Table 8: Intermediate training results of Qwen-3-8B-Base on DeepMath-6K with PPO and SPEC-RL. We report rollout efficiency and accuracy every 10 training steps, with PPO and its SPEC-RL variant interleaved.

| Algorithm | Step | Rollout Efficiency | | Math Reasoning | | | | | OOD | | AVG |
| | | Tokens (M) | Speedup | AMC23 | GSM8K | MATH 500 | Minerva Math | Olympiad Bench | MMLU STEM | IFEval | |
|---|---|---|---|---|---|---|---|---|---|---|---|
| Base Model | 0 | - | - | 40.0 | 83.0 | 67.4 | 27.2 | 34.1 | 60.4 | 29.9 | 48.9 |
| PPO | 10 | 73.4 | 1.00× | 42.5 | 91.1 | 75.4 | 33.5 | 43.4 | 63.2 | 32.2 | 54.5 |
| ↪ + SPEC-RL | 10 | 51.9 | 1.30× | 50.0 | 92.1 | 79.6 | 32.0 | 42.4 | 62.6 | 36.0 | 56.4 |
| PPO | 20 | 144.8 | 1.00× | 60.0 | 93.1 | 81.0 | 39.3 | 45.3 | 67.0 | 35.1 | 60.1 |
| ↪ + SPEC-RL | 20 | 85.6 | 1.48× | 52.5 | 93.3 | 82.0 | 39.7 | 45.5 | 67.4 | 37.9 | 59.8 |
| PPO | 30 | 241.0 | 1.00× | 62.5 | 93.4 | 82.8 | 38.6 | 46.7 | 77.6 | 37.3 | 62.7 |
| ↪ + SPEC-RL | 30 | 115.6 | 1.75× | 60.0 | 92.6 | 82.2 | 40.4 | 48.1 | 74.6 | 42.7 | 62.9 |
| PPO | 40 | 359.0 | 1.00× | 60.0 | 92.9 | 83.6 | 41.9 | 49.3 | 79.5 | 36.6 | 63.4 |
| ↪ + SPEC-RL | 40 | 159.4 | 1.87× | 62.5 | 94.0 | 84.0 | 39.3 | 49.5 | 77.6 | 43.3 | 64.3 |
| PPO | 50 | 484.2 | 1.00× | 65.0 | 93.5 | 86.0 | 41.2 | 51.6 | 82.1 | 39.7 | 65.6 |
| ↪ + SPEC-RL | 50 | 197.9 | 1.98× | 67.5 | 93.1 | 84.2 | 42.6 | 49.3 | 81.9 | 41.8 | 65.8 |
| PPO | 60 | 609.9 | 1.00× | 75.0 | 94.2 | 85.4 | 42.6 | 49.9 | 82.9 | 42.0 | 67.4 |
| ↪ + SPEC-RL | 60 | 251.2 | 1.95× | 67.5 | 93.3 | 84.6 | 43.8 | 52.0 | 81.7 | 43.6 | 66.6 |
| PPO | 70 | 735.0 | 1.00× | 82.5 | 93.5 | 84.4 | 44.1 | 51.1 | 83.6 | 42.9 | 68.9 |
| ↪ + SPEC-RL | 70 | 307.9 | 1.91× | 70.0 | 93.7 | 84.6 | 42.6 | 50.8 | 84.0 | 44.7 | 67.2 |
| PPO | 80 | 859.9 | 1.00× | 62.5 | 93.8 | 85.6 | 42.3 | 51.9 | 83.3 | 40.9 | 65.8 |
| ↪ + SPEC-RL | 80 | 358.1 | 1.90× | 75.0 | 93.5 | 83.4 | 44.1 | 50.2 | 84.4 | 43.1 | 67.7 |
| PPO | 90 | 984.0 | 1.00× | 70.0 | 94.2 | 85.8 | 43.0 | 51.6 | 83.8 | 41.6 | 67.1 |
| ↪ + SPEC-RL | 90 | 400.1 | 1.94× | 75.0 | 92.9 | 85.2 | 43.4 | 50.8 | 84.4 | 41.0 | 67.5 |

Table 9: Intermediate training results of LLaMA-3.2-1B-Instruct on DeepMath-6K with PPO and SPEC-RL. We report rollout efficiency and accuracy every 10 training steps, with PPO and its SPEC-RL variant interleaved.

| Algorithm | Step | Rollout Efficiency | | Math Reasoning | | | | | OOD | | AVG |
| | | Tokens (M) | Speedup | AMC23 | GSM8K | MATH 500 | Minerva Math | Olympiad Bench | MMLU STEM | IFEval | |
|---|---|---|---|---|---|---|---|---|---|---|---|
| Base Model | 0 | - | - | 0.0 | 26.7 | 14.2 | 4.0 | 2.8 | 32.6 | 37.0 | 16.8 |
| PPO | 10 | 65.5 | 1.00× | 2.5 | 26.5 | 11.4 | 4.0 | 3.6 | 33.0 | 42.0 | 17.6 |
| ↪ + SPEC-RL | 10 | 50.2 | 1.26× | 20.0 | 25.8 | 14.0 | 2.6 | 3.7 | 32.5 | 41.0 | 19.9 |
| PPO | 20 | 131.1 | 1.00× | 7.5 | 26.7 | 16.2 | 3.7 | 4.7 | 33.9 | 35.9 | 18.4 |
| ↪ + SPEC-RL | 20 | 89.0 | 1.35× | 7.5 | 27.9 | 16.2 | 2.9 | 5.0 | 34.1 | 41.2 | 19.3 |
| PPO | 30 | 192.0 | 1.00× | 12.5 | 28.8 | 17.2 | 3.7 | 4.1 | 35.3 | 38.6 | 20.0 |
| ↪ + SPEC-RL | 30 | 118.3 | 1.47× | 10.0 | 29.8 | 17.4 | 4.4 | 6.4 | 34.2 | 38.8 | 20.1 |
| PPO | 40 | 250.3 | 1.00× | 5.0 | 29.7 | 19.6 | 2.9 | 4.4 | 35.1 | 39.0 | 19.4 |
| ↪ + SPEC-RL | 40 | 134.8 | 1.63× | 15.0 | 31.6 | 18.6 | 3.3 | 6.1 | 33.2 | 39.9 | 21.1 |
| PPO | 50 | 306.8 | 1.00× | 5.0 | 31.3 | 19.2 | 4.8 | 4.6 | 32.3 | 40.1 | 19.6 |
| ↪ + SPEC-RL | 50 | 147.2 | 1.78× | 10.0 | 31.9 | 19.4 | 5.1 | 5.0 | 35.3 | 40.3 | 21.0 |
| PPO | 60 | 361.6 | 1.00× | 7.5 | 30.3 | 18.4 | 5.1 | 4.3 | 35.1 | 41.6 | 20.3 |
| ↪ + SPEC-RL | 60 | 160.5 | 1.89× | 12.5 | 31.5 | 19.2 | 4.8 | 5.5 | 34.4 | 39.9 | 21.1 |
| PPO | 70 | 415.4 | 1.00× | 12.5 | 31.0 | 17.8 | 3.7 | 6.1 | 35.1 | 40.1 | 20.9 |
| ↪ + SPEC-RL | 70 | 175.8 | 1.95× | 12.5 | 32.6 | 19.6 | 3.3 | 5.5 | 34.9 | 39.9 | 21.2 |
| PPO | 80 | 469.0 | 1.00× | 10.0 | 34.1 | 19.6 | 4.8 | 4.1 | 34.8 | 41.6 | 21.3 |
| ↪ + SPEC-RL | 80 | 188.8 | 2.02× | 15.0 | 33.5 | 19.0 | 5.5 | 6.4 | 36.1 | 40.3 | 22.3 |
| PPO | 90 | 521.5 | 1.00× | 10.0 | 31.6 | 20.8 | 4.0 | 6.4 | 34.3 | 42.7 | 21.4 |
| ↪ + SPEC-RL | 90 | 210.6 | 2.01× | 10.0 | 32.4 | 20.2 | 5.5 | 5.0 | 35.3 | 40.7 | 21.3 |

Table 10: Intermediate training results of Qwen-3-1.7B-Base on DeepMath-6K with DAPO and SPEC-RL. Since DAPO adopts *Dynamic Sampling*, one training step may correspond to multiple generation steps; thus we additionally report the **Gen-Step** column to indicate how many rollout batches the model has consumed.

| Algorithm | Step | Gen-Step | Rollout Efficiency | | Math Reasoning | | | | | OOD | | AVG |
|---|---|---|---|---|---|---|---|---|---|---|---|---|
| | | | Tokens (M) | Speedup | AMC23 | GSM8K | MATH 500 | Minerva Math | Olympiad Bench | MMLU STEM | IFEval | |
| Base Model | 0 | 0 | - | - | 22.5 | 59.1 | 45.0 | 12.5 | 16.7 | 39.3 | 17.9 | 30.4 |
| DAPO | 5 | 10 | 65.2 | 1.00× | 25.0 | 69.8 | 50.0 | 17.6 | 20.0 | 41.5 | 20.1 | 34.9 |
| ↪ + SPEC-RL | 5 | 10 | 46.7 | 1.25× | 35.0 | 69.4 | 53.4 | 20.2 | 21.0 | 43.0 | 17.7 | 37.1 |
| DAPO | 10 | 20 | 127.4 | 1.00× | 32.5 | 76.0 | 58.4 | 19.1 | 22.4 | 42.9 | 19.6 | 38.7 |
| ↪ + SPEC-RL | 10 | 20 | 70.6 | 1.52× | 15.0 | 76.3 | 56.8 | 16.5 | 20.7 | 44.1 | 21.6 | 35.9 |
| DAPO | 15 | 30 | 187.7 | 1.00× | 27.5 | 76.5 | 56.4 | 19.9 | 22.7 | 45.8 | 20.7 | 38.5 |
| ↪ + SPEC-RL | 15 | 30 | 95.5 | 1.60× | 25.0 | 78.2 | 58.4 | 23.9 | 24.9 | 45.0 | 26.1 | 40.2 |
| DAPO | 20 | 40 | 247.6 | 1.00× | 35.0 | 78.0 | 54.8 | 23.2 | 21.8 | 46.8 | 23.3 | 40.4 |
| ↪ + SPEC-RL | 20 | 40 | 109.5 | 1.74× | 32.5 | 78.5 | 57.2 | 25.0 | 24.9 | 47.7 | 24.6 | 41.5 |
| DAPO | 25 | 50 | 307.5 | 1.00× | 35.0 | 77.2 | 59.4 | 20.6 | 25.9 | 47.2 | 19.4 | 40.7 |
| ↪ + SPEC-RL | 25 | 50 | 124.9 | 1.84× | 37.5 | 77.8 | 59.0 | 22.8 | 23.0 | 49.2 | 25.7 | 42.1 |
| DAPO | 30 | 60 | 367.0 | 1.00× | 35.0 | 79.1 | 60.6 | 25.0 | 24.3 | 48.0 | 22.2 | 42.0 |
| ↪ + SPEC-RL | 30 | 60 | 137.7 | 1.94× | 32.5 | 79.4 | 60.4 | 24.6 | 25.0 | 50.4 | 26.2 | 42.6 |
| DAPO | 35 | 70 | 425.6 | 1.00× | 37.5 | 78.5 | 59.8 | 27.9 | 24.3 | 49.7 | 22.7 | 42.9 |
| ↪ + SPEC-RL | 35 | 70 | 149.1 | 2.02× | 30.0 | 80.3 | 62.2 | 25.0 | 25.3 | 51.6 | 25.0 | 42.8 |
| DAPO | 40 | 80 | 484.6 | 1.00× | 27.5 | 79.8 | 61.6 | 24.6 | 25.0 | 50.8 | 22.7 | 41.7 |
| ↪ + SPEC-RL | 40 | 80 | 160.2 | 2.10× | 40.0 | 79.2 | 60.2 | 25.4 | 26.5 | 53.7 | 27.4 | 44.6 |
| DAPO | 45 | 90 | 543.1 | 1.00× | 30.0 | 79.6 | 60.8 | 24.6 | 23.0 | 52.2 | 24.8 | 42.1 |
| ↪ + SPEC-RL | 45 | 90 | 171.6 | 2.17× | 22.5 | 80.1 | 60.0 | 25.7 | 25.5 | 53.5 | 27.0 | 42.0 |

Table 11: Intermediate training results of Qwen-3-8B-Base on DeepMath-6K with DAPO and SPEC-RL. Since DAPO adopts *Dynamic Sampling*, one training step may correspond to multiple generation steps; thus we additionally report the **Gen-Step** column to indicate how many rollout batches the model has consumed.

| Algorithm | Step | Gen-Step | Rollout Efficiency | | Math Reasoning | | | | | OOD | | AVG |
|---|---|---|---|---|---|---|---|---|---|---|---|---|
| | | | Tokens (M) | Speedup | AMC23 | GSM8K | MATH 500 | Minerva Math | Olympiad Bench | MMLU STEM | IFEval | |
| *Qwen-3-8B-DAPO-SPEC-RL* | | | | | | | | | | | | |
| Base Model | 0 | 0 | - | - | 40.0 | 83.0 | 67.4 | 27.2 | 34.1 | 60.4 | 29.9 | 48.9 |
| DAPO | 5 | 10 | 75.0 | 1.00× | 62.5 | 89.8 | 73.6 | 27.9 | 39.1 | 60.5 | 32.0 | 55.1 |
| ↪ + SPEC-RL | 5 | 10 | 59.0 | 1.20× | 55.0 | 90.9 | 75.0 | 32.7 | 38.8 | 63.3 | 34.0 | 55.7 |
| DAPO | 10 | 20 | 148.8 | 1.00× | 60.0 | 91.9 | 78.6 | 36.8 | 43.4 | 64.0 | 36.0 | 58.7 |
| ↪ + SPEC-RL | 10 | 20 | 90.3 | 1.45× | 60.0 | 92.8 | 79.0 | 37.1 | 40.7 | 63.0 | 34.2 | 58.1 |
| DAPO | 15 | 30 | 235.5 | 1.00× | 67.5 | 93.3 | 80.6 | 39.7 | 47.4 | 70.6 | 38.1 | 62.5 |
| ↪ + SPEC-RL | 15 | 30 | 116.8 | 1.73× | 60.0 | 91.9 | 81.8 | 42.6 | 47.9 | 69.6 | 39.0 | 61.8 |
| DAPO | 20 | 40 | 354.9 | 1.00× | 62.5 | 93.1 | 84.6 | 41.2 | 46.1 | 77.5 | 36.4 | 63.1 |
| ↪ + SPEC-RL | 20 | 40 | 152.2 | 2.00× | 70.0 | 93.5 | 83.8 | 39.0 | 49.8 | 75.4 | 38.1 | 64.2 |
| DAPO | 25 | 51 | 509.1 | 1.00× | 62.5 | 93.1 | 83.4 | 39.3 | 49.6 | 79.8 | 38.8 | 63.8 |
| ↪ + SPEC-RL | 25 | 50 | 199.8 | 2.19× | 72.5 | 92.6 | 85.4 | 41.5 | 47.9 | 78.7 | 42.1 | 65.8 |
| DAPO | 30 | 63 | 685.0 | 1.00× | 62.5 | 92.5 | 83.8 | 44.5 | 48.9 | 81.1 | 39.2 | 64.6 |
| ↪ + SPEC-RL | 30 | 60 | 239.9 | 2.48× | 70.0 | 93.9 | 84.0 | 39.7 | 48.7 | 80.2 | 40.3 | 65.3 |
| DAPO | 35 | 75 | 867.6 | 1.00× | 75.0 | 92.6 | 82.8 | 40.8 | 49.5 | 81.9 | 38.1 | 65.8 |
| ↪ + SPEC-RL | 35 | 70 | 278.9 | 2.73× | 72.5 | 93.6 | 84.8 | 40.4 | 49.9 | 80.8 | 44.4 | 66.6 |
| DAPO | 40 | 87 | 1052.2 | 1.00× | 75.0 | 93.3 | 84.8 | 40.1 | 48.6 | 82.4 | 39.6 | 66.3 |
| ↪ + SPEC-RL | 40 | 82 | 326.2 | 2.88× | 65.0 | 93.8 | 84.4 | 43.8 | 50.4 | 82.2 | 44.4 | 66.3 |

Table 12: Intermediate training results of LLaMA-3.2-1B-Instruct on DeepMath-6K with DAPO and SPEC-RL. Since DAPO adopts *Dynamic Sampling*, one training step may correspond to multiple generation steps; thus we additionally report the **Gen-Step** column to indicate how many rollout batches the model has consumed.

| Algorithm | Step | Gen-Step | Rollout Efficiency | | Math Reasoning | | | | | OOD | | AVG |
|---|---|---|---|---|---|---|---|---|---|---|---|---|
| | | | Tokens (M) | Speedup | AMC23 | GSM8K | MATH 500 | Minerva Math | Olympiad Bench | MMLU STEM | IFEval | |
| *LLaMA-3.2-1B-DAPO-SPEC-RL* | | | | | | | | | | | | |
| Base Model | 0 | 0 | - | - | 0.0 | 26.7 | 14.2 | 4.0 | 2.8 | 32.6 | 37.0 | 16.8 |
| DAPO | 5 | 15 | 105.6 | 1.00× | 2.5 | 27.1 | 14.4 | 2.9 | 3.3 | 32.6 | 38.8 | 17.4 |
| ↪ + SPEC-RL | 5 | 15 | 52.4 | 1.95× | 7.5 | 27.0 | 14.4 | 2.6 | 3.0 | 32.7 | 38.3 | 17.9 |
| DAPO | 10 | 27 | 179.8 | 1.00× | 5.0 | 25.5 | 14.6 | 2.6 | 4.1 | 34.8 | 40.3 | 18.1 |
| ↪ + SPEC-RL | 10 | 28 | 69.2 | 2.16× | 17.5 | 25.6 | 16.6 | 2.6 | 4.9 | 33.7 | 39.2 | 20.0 |
| DAPO | 15 | 38 | 239.8 | 1.00× | 5.0 | 27.1 | 18.4 | 4.4 | 4.4 | 33.9 | 37.3 | 18.6 |
| ↪ + SPEC-RL | 15 | 39 | 79.2 | 2.19× | 5.0 | 28.9 | 16.4 | 6.2 | 3.4 | 33.9 | 37.5 | 18.8 |
| DAPO | 20 | 53 | 322.1 | 1.00× | 17.5 | 27.3 | 18.8 | 2.9 | 5.6 | 34.6 | 38.8 | 20.8 |
| ↪ + SPEC-RL | 20 | 53 | 92.4 | 2.31× | 7.5 | 29.9 | 19.6 | 5.9 | 5.5 | 34.2 | 38.1 | 20.1 |
| DAPO | 25 | 68 | 402.9 | 1.00× | 5.0 | 26.6 | 19.8 | 3.3 | 4.6 | 34.5 | 38.3 | 18.9 |
| ↪ + SPEC-RL | 25 | 68 | 105.3 | 2.43× | 10.0 | 34.0 | 19.8 | 4.0 | 6.1 | 35.5 | 35.5 | 20.7 |
| DAPO | 30 | 83 | 482.6 | 1.00× | 7.5 | 29.6 | 19.2 | 4.0 | 5.5 | 33.0 | 38.6 | 19.6 |
| ↪ + SPEC-RL | 30 | 83 | 123.1 | 2.48× | 10.0 | 34.9 | 20.2 | 4.0 | 5.5 | 35.5 | 38.4 | 21.2 |

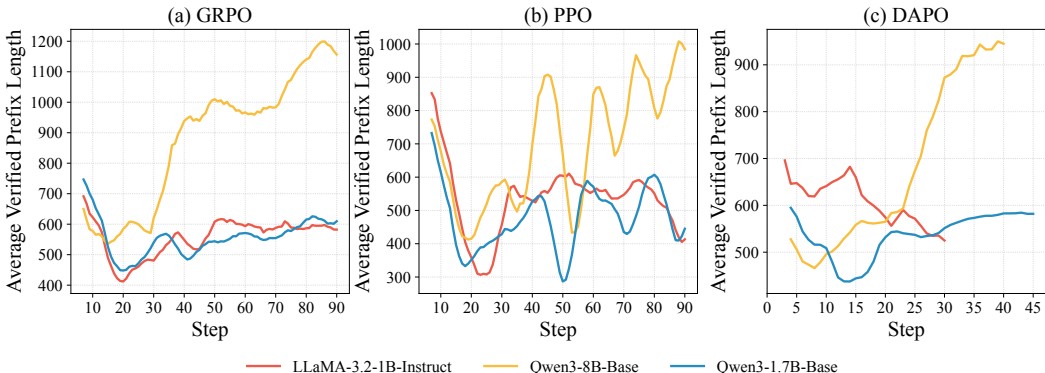

Figure 7: Average verified prefix length trajectories of SPEC-RL under three RL algorithms: (a) GRPO, (b) PPO, and (c) DAPO. The y-axis reports the average length of the verified speculative prefix per training step, and the x-axis is the training step. colors denote model backbones: red: LLaMA-3.2-1B-Instruct, yellow: Qwen3-8B-Base, blue: Qwen3-1.7B-Base.

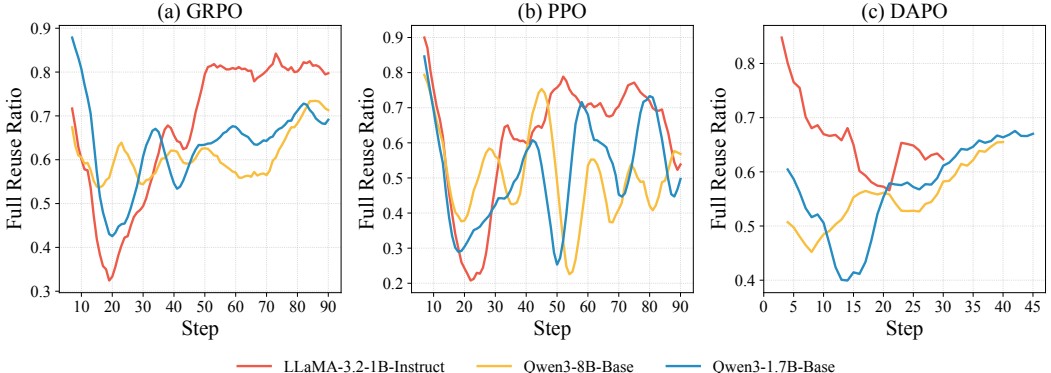

Figure 8: Full reuse ratio trajectories of SPEC-RL under three RL algorithms: (a) GRPO, (b) PPO, and (c) DAPO. The y-axis reports the fraction of rollouts per step that are fully reused, and the x-axis the training step. colors denote model backbones: red: LLaMA-3.2-1B-Instruct, yellow: Qwen3-8B-Base, blue: Qwen3-1.7B-Base. Across settings, SPEC-RL quickly stabilizes at a high full reuse ratio, indicating effective speculative reuse during training.

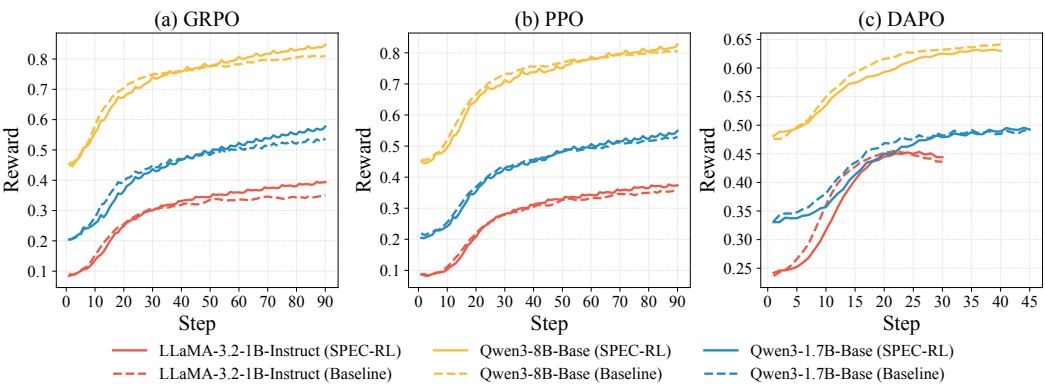

Figure 9: Training reward trajectories of SPEC-RL versus baseline under three RL algorithms: (a) GRPO, (b) PPO, and (c) DAPO. The y-axis reports reward, and the x-axis the training step. colors denote model backbones: red: LLaMA-3.2-1B-Instruct, yellow: Qwen3-8B-Base, blue: Qwen3-1.7B-Base, while solid lines indicate SPEC-RL and dashed lines the corresponding vanilla baselines. SPEC-RL matches or exceeds baseline rewards under different algorithms across all backbones.

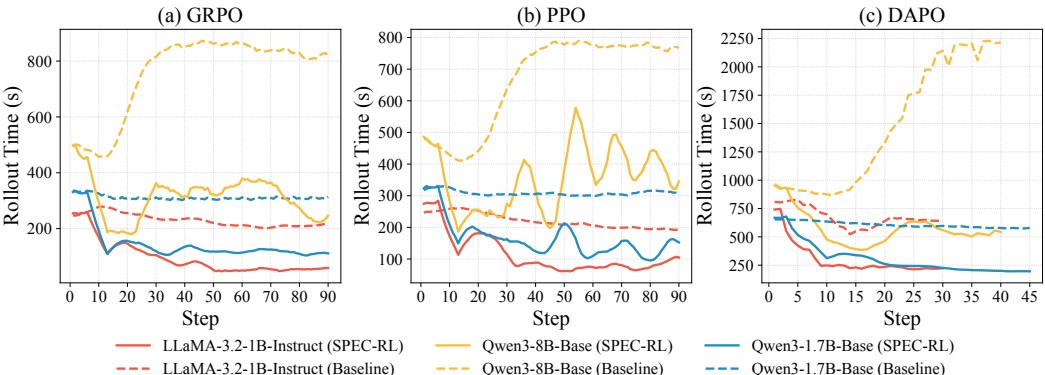

Figure 10: Training rollout time of SPEC-RL versus baseline under three RL algorithms: (a) GRPO, (b) PPO, and (c) DAPO. The y-axis reports rollout time (seconds) and the x-axis the training step. colors denote model backbones: red: LLaMA-3.2-1B-Instruct, yellow: Qwen3-8B-Base, blue: Qwen3-1.7B-Base, while solid lines indicate SPEC-RL and dashed lines the corresponding vanilla baselines. Across algorithms and models, SPEC-RL yields consistently lower rollout time than the baselines.

Table 13: End-to-end training time comparison across models and algorithms. We report both the **wall-clock training hours** ("End-to-end (h)") and the **average step time** ("Total (s)") with a detailed breakdown. *validation* refers to our newly introduced speculative decoding process that verifies old-policy rollouts in parallel; *assemble* denotes combining verified prefixes with newly generated continuations to form complete rollouts; the remaining parts (*reward*, *old-log-probs*, *ref*, *values*, *adv*, *update-critic*, *update-actor*, *others*) follow the standard pipeline of the verl framework in execution order.

| Algorithm | End-to-end (h) Total | Total | Δ vs. base | verification | rollout | assembly | reward | old-log-probs | ref | values | adv | update-critic | update-actor | others |
|---|---|---|---|---|---|---|---|---|---|---|---|---|---|---|
| | | | | | *Qwen-3-1.7B-Base* | | | | | | | | | |
| GRPO | 12.63 | 505.1 | – | – | 309.9 | – | 91.0 | 17.2 | 15.8 | – | 0.4 | – | 56.0 | 14.9 |
| ↪ + SPEC-RL | 8.65 | 346.0 | ↓ 159.1 | 22.1 | 135.2 (2.29×) | 1.5 | 81.0 | 17.1 | 16.3 | – | 0.5 | – | 56.2 | 16.2 |
| PPO | 14.10 | 563.9 | – | – | 308.1 | – | 100.5 | 17.2 | – | 14.0 | 4.7 | 46.0 | 56.5 | 16.9 |
| ↪ + SPEC-RL | 10.78 | 431.2 | ↓ 132.7 | 22.7 | 158.6 (1.94×) | 1.4 | 94.1 | 17.3 | – | 13.8 | 4.6 | 45.0 | 55.5 | 18.1 |
| DAPO | 11.10 | 443.8 | – | – | 301.3 | – | 93.1 | 8.6 | – | – | 0.3 | – | 25.9 | 14.6 |
| ↪ + SPEC-RL | 7.90 | 316.0 | ↓ 127.9 | 21.0 | 139.0 (2.17×) | 1.4 | 97.9 | 18.1 | – | – | 0.2 | – | 25.9 | 12.7 |
| | | | | | *Qwen-3-8B-Base* | | | | | | | | | |
| GRPO | 31.66 | 1266.4 | – | – | 768.2 | – | 73.2 | 66.8 | 66.9 | – | 4.2 | – | 263.8 | 23.4 |
| ↪ + SPEC-RL | 21.03 | 841.0 | ↓ 425.4 | 74.7 | 305.8 (2.51×) | 1.3 | 61.4 | 63.8 | 62.4 | – | 4.9 | – | 248.8 | 18.0 |
| PPO | 34.85 | 1393.9 | – | – | 676.7 | – | 70.5 | 65.4 | – | 57.4 | 4.2 | 224.1 | 260.4 | 35.3 |
| ↪ + SPEC-RL | 26.97 | 1078.8 | ↓ 315.1 | 71.5 | 349.3 (1.94×) | 1.4 | 64.9 | 59.6 | – | 52.1 | 4.9 | 205.9 | 236.9 | 32.5 |
| DAPO | 24.29 | 971.8 | – | – | 699.2 | – | 64.4 | 66.3 | – | – | 0.1 | – | 121.1 | 20.7 |
| ↪ + SPEC-RL | 12.90 | 515.9 | ↓ 455.9 | 51.0 | 243.0 (2.88×) | 1.1 | 54.0 | 51.2 | – | – | 0.1 | – | 97.5 | 18.0 |
| | | | | | *LLaMA-3.2-1B-Instruct* | | | | | | | | | |
| GRPO | 10.20 | 408.0 | – | – | 229.7 | – | 105.8 | 12.6 | 11.5 | – | 0.4 | – | 34.7 | 13.2 |
| ↪ + SPEC-RL | 7.28 | 291.3 | ↓ 116.7 | 17.2 | 88.3 (2.60×) | 1.4 | 110.4 | 13.0 | 11.9 | – | 0.5 | – | 34.4 | 14.4 |
| PPO | 10.94 | 437.6 | – | – | 218.9 | – | 117.6 | 12.5 | – | 10.0 | 4.8 | 10.0 | 32.6 | 31.3 |
| ↪ + SPEC-RL | 8.60 | 344.0 | ↓ 93.6 | 17.5 | 108.9 (2.01×) | 1.3 | 110.9 | 12.4 | – | 10.1 | 4.6 | 10.1 | 34.3 | 33.8 |
| DAPO | 9.77 | 328.4 | – | – | 198.4 | – | 100.8 | 11.2 | – | – | 0.1 | – | 9.6 | 8.5 |
| ↪ + SPEC-RL | 6.97 | 238.4 | ↓ 90.0 | 13.4 | 80.0 (2.48×) | 1.1 | 110.5 | 11.5 | – | – | 0.1 | – | 9.9 | 12.0 |

## A.4 END-TO-END TIME BREAKDOWN

Table 13 reports the per-stage breakdown of training time. In the vanilla baseline, rollout generation dominates the runtime, often accounting for more than 60% of the total. With SPEC-RL, this cost is largely shifted into a lightweight verification stage, where cached rollouts are first verified in parallel under the current policy and then evaluated by the speculative decoding rule to determine the rejection position, and a minimal assembly stage, where verified prefixes and regenerated suffixes are merged into complete responses. Both stages add only minor overhead (on Qwen-3-1.7B-Base, verification ∼20s and assembly ∼1–2s), while the total step time is reduced by about 129–161s, making the extra cost negligible compared to the savings from reduced rollout. For instance, on Qwen-3-8B-Base/GRPO, the rollout time decreases from 768.2s to 305.8s, while all other stages, such as reward computation and policy updates, remain nearly unchanged. Overall, although these new stages slightly increase non-rollout costs, the dominant effect is the 2–3 times reduction in rollout tokens, yielding substantially faster end-to-end training.

## A.5 GENERALITY ACROSS DATASETS

To examine whether the gains of SPEC-RL depend on a specific training corpus, we conduct experiments on two distinct datasets: DeepMath-6K and SimpleRL-8K. Results in Table 14 show that SPEC-RL consistently improves rollout efficiency across both settings. For example, on Qwen-3-1.7B-Base with GRPO, rollout tokens drop from 554.8M to 182.7M on DeepMath-6K and from 639.4M to 354.0M on SimpleRL-8K. Accuracy remains comparable or slightly improved, confirming that the efficiency benefits of SPEC-RL are robust to the choice of dataset. Intermediate perfor-

Table 14: Ablation study on different training datasets. Results show that our method maintains improvements in rollout efficiency and accuracy across both Deepmath-6K and SimpleRL-8K settings.

| Algorithm | Rollout Efficiency | | Math Reasoning | | | | | OOD | | AVG |
|---|---|---|---|---|---|---|---|---|---|---|
| | Tokens (M) | Speedup | AMC23 | GSM8K | MATH 500 | Minerva Math | Olympiad Bench | MMLU STEM | IFEval | |
| | | | *Deepmath-6K (Qwen-3-1.7B-Base)* | | | | | | | |
| GRPO | 554.8 | 1.00× | **42.5** | 82.6 | 64.4 | 26.5 | 25.5 | **60.7** | 24.4 | 46.7 |
| ↪ + SPEC-RL | **182.7** | **2.29×** | 37.5 | **84.4** | **68.0** | **29.4** | **29.3** | 58.3 | **28.8** | **48.0** |
| | | | *SimpleRL-8K (Qwen-3-1.7B-Base)* | | | | | | | |
| GRPO | 639.4 | 1.00× | **45.0** | 83.8 | 68.2 | **27.2** | 30.5 | 49.4 | 24.0 | 46.9 |
| ↪ + SPEC-RL | **354.0** | **1.53×** | 40.0 | **85.1** | **72.2** | **27.2** | **32.1** | 57.4 | **27.7** | **48.8** |

Table 15: Intermediate training results of Qwen-3-1.7B-Base on SimpleRL-8K with GRPO and SPEC-RL. We report rollout efficiency and accuracy every 10 training steps, with GRPO and its SPEC-RL variant interleaved.

| Algorithm | Step | Rollout Efficiency | | Math Reasoning | | | | | OOD | | AVG |
| | | Tokens (M) | Speedup | AMC23 | GSM8K | MATH 500 | Minerva Math | Olympiad Bench | MMLU STEM | IFEval | |
| --- | --- | --- | --- | --- | --- | --- | --- | --- | --- | --- | --- |
| Base Model | 0 | - | - | 22.5 | 59.1 | 45.0 | 12.5 | 16.7 | 39.3 | 17.9 | 30.4 |
| GRPO | 10 | 61.5 | 1.00× | 35.0 | 76.7 | 57.6 | 19.1 | 22.7 | 44.4 | 21.1 | 39.5 |
| ↪ + SPEC-RL | 10 | 50.5 | 1.16× | 25.0 | 75.9 | 57.8 | 19.1 | 24.9 | 44.9 | 21.3 | 38.4 |
| GRPO | 20 | 123.1 | 1.00× | 27.5 | 78.5 | 60.6 | 22.1 | 25.9 | 46.1 | 23.7 | 40.6 |
| ↪ + SPEC-RL | 20 | 84.3 | 1.33× | 25.0 | 79.3 | 62.2 | 20.6 | 28.0 | 46.4 | 20.9 | 40.3 |
| GRPO | 30 | 185.3 | 1.00× | 45.0 | 80.2 | 63.8 | 24.3 | 27.0 | 46.1 | 23.7 | 44.3 |
| ↪ + SPEC-RL | 30 | 112.2 | 1.47× | 45.0 | 81.5 | 61.4 | 26.8 | 28.7 | 49.0 | 25.1 | 45.4 |
| GRPO | 40 | 247.7 | 1.00× | 32.5 | 80.1 | 63.0 | 22.4 | 28.1 | 46.9 | 21.1 | 42.0 |
| ↪ + SPEC-RL | 40 | 136.8 | 1.57× | 37.5 | 81.3 | 65.8 | 24.3 | 29.5 | 49.8 | 26.1 | 44.9 |
| GRPO | 50 | 312.5 | 1.00× | 37.5 | 79.7 | 65.2 | 27.2 | 26.8 | 48.7 | 23.1 | 44.0 |
| ↪ + SPEC-RL | 50 | 171.4 | 1.58× | 35.0 | 83.9 | 66.2 | 28.7 | 30.8 | 52.9 | 26.2 | 46.2 |
| GRPO | 60 | 377.7 | 1.00× | 40.0 | 82.0 | 64.6 | 26.5 | 28.0 | 48.5 | 23.3 | 44.7 |
| ↪ + SPEC-RL | 60 | 206.9 | 1.58× | 47.5 | 83.1 | 67.8 | 26.8 | 32.0 | 53.5 | 25.5 | 48.0 |
| GRPO | 70 | 444.8 | 1.00× | 37.5 | 81.7 | 66.0 | 26.5 | 26.5 | 48.6 | 20.9 | 44.0 |
| ↪ + SPEC-RL | 70 | 246.5 | 1.56× | 47.5 | 83.2 | 70.0 | 28.3 | 31.3 | 53.8 | 27.9 | 48.9 |
| GRPO | 80 | 512.9 | 1.00× | 45.0 | 82.3 | 66.8 | 26.5 | 30.5 | 47.5 | 25.0 | 46.2 |
| ↪ + SPEC-RL | 80 | 283.2 | 1.55× | 47.5 | 83.9 | 68.4 | 26.5 | 31.7 | 55.5 | 25.0 | 48.4 |
| GRPO | 90 | 582.6 | 1.00× | 42.5 | 83.2 | 66.6 | 25.7 | 29.5 | 48.4 | 24.6 | 45.8 |
| ↪ + SPEC-RL | 90 | 324.8 | 1.53× | 35.0 | 83.5 | 70.4 | 27.9 | 31.9 | 55.4 | 27.0 | 47.3 |
| GRPO | 100 | 639.4 | 1.00× | 45.0 | 83.8 | 68.2 | 27.2 | 30.5 | 49.4 | 24.0 | 46.9 |
| ↪ + SPEC-RL | 100 | 354.0 | 1.54× | 40.0 | 85.1 | 72.2 | 27.2 | 32.1 | 57.4 | 27.7 | 48.8 |

mance on SimpleRL-8K is reported in Table 15, while the detailed results for DeepMath-6K can be found in Table 4. These results suggest that the efficiency improvements of SPEC-RL do not rely on a particular training distribution.

## A.6 Impact of Training Set Size on Acceleration

Since SPEC-RL accelerates training by reusing cached rollouts from the previous epoch, acceleration can only take effect starting from the second epoch. To study how dataset size influences this effect, we vary the training set size to 2K, 3K, 4K, 5K, and 6K samples, and train Qwen-3-1.7B-Base with GRPO. Figure 11 reports the rollout time across training steps.

We observe that smaller datasets lead to earlier reuse opportunities, since epochs finish more quickly and the second epoch arrives sooner. For example, with 2K samples, the rollout time drops sharply after step 3, whereas with 6K samples, the reduction is delayed until later steps. Across all settings, rollout time decreases steadily once reuse begins, with larger speedups achieved as training progresses. The markers in the figure denote the first reuse points (the first step of epoch 2), where SPEC-RL begins to take effect. This analysis confirms that the efficiency gains of SPEC-RL depend not only on algorithm and model choice, but also on the dataset size, which determines how soon reuse can be activated during training.

## A.7 Random Reuse Results

For completeness, we also report the full training trajectories of the Random Reuse baseline on Qwen-3-1.7B-Base, trained with GRPO on DeepMath-6K.

Table 16 interleaves results of GRPO and Random Reuse every 10 training steps, providing a stepwise view of rollout efficiency and accuracy. While Table 2 summarizes the overall comparison, these detailed results illustrate how Random Reuse accelerates rollouts but produces unstable performance over the course of training.

## A.8 Full Lenience Ablation Results

For completeness, we provide the step-level results corresponding to the lenience ablation in Section 4.3. While Table 3 reports only the final-step outcomes for comparison across different lenience values, we include detailed intermediate results every 10 training steps here. These tables document how rollout efficiency and accuracy evolve throughout training under various lenience

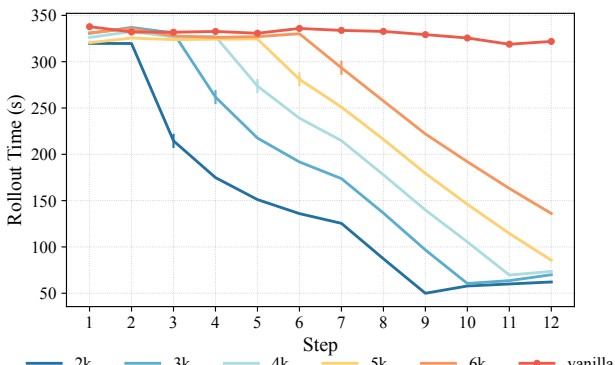

Figure 11: Rollout time under different training set sizes (2K–6K) with GRPO on Qwen-3-1.7B-Base. Markers highlight the first reuse points at the start of epoch 2, when SPEC-RL begins to accelerate rollouts.

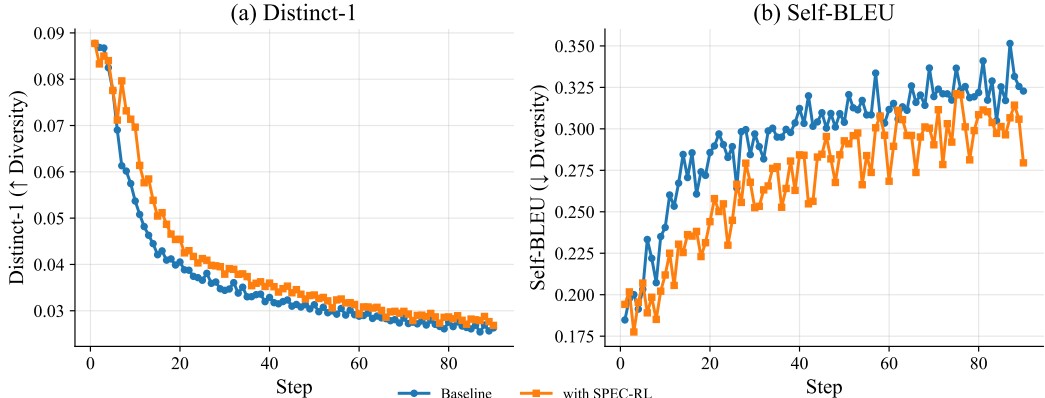

Figure 12: **Effects of SPEC-RL on rollout diversity.** We compare GRPO and SPEC-RL using two standard diversity metrics: **Distinct-1**, where higher values indicate greater unigram diversity, and **Self-BLEU**, where lower values indicate greater sample diversity. Across identical training steps, SPEC-RL attains diversity levels comparable to—or slightly higher than—the GRPO baseline.

settings ($\ell = 1, e^{0.2}, e^{0.5}, e^{0.8}, e^{1.0}, e^{2.0}$, and $\ell \to \infty$), complementing the aggregated trends shown in Table 3.

## A.9 ANALYSIS OF ROLLOUT DIVERSITY

We further examine whether speculative reuse affects rollout diversity by comparing SPEC-RL with the GRPO baseline at identical training steps. Diversity is measured using two standard metrics: **distinct-1** (Li et al., 2016), which captures unigram variability, and **Self-BLEU** (Zhu et al., 2018), which measures sample similarity within a batch. As shown in Figure 12(a)–(b), SPEC-RL consistently achieves *equal or even higher* diversity than the GRPO baseline throughout training. Although SPEC-RL reuses prefixes generated by the previous policy, this reuse only makes the prefix region more similar to the old policy's outputs; it does *not* constrain the diversity of the current policy's rollouts produced at the same step, compared with standard GRPO. In practice, the variation across trajectories at each step—the quantity that matters for exploration and effective learning—remains fully preserved. Importantly, there is no empirical evidence that higher similarity between the current policy and the old policy reduces trajectory-level diversity or negatively affects optimization. The observed trends confirm that SPEC-RL maintains, and in some cases slightly improves, rollout diversity without harming learning dynamics or final performance.

Table 16: Intermediate training results of Qwen-3-1.7B-Base on DeepMath-6K with GRPO and Random Reuse. We report rollout efficiency and accuracy every 10 training steps, with GRPO and its Random Reuse variant interleaved.

| Algorithm | Step | Rollout Efficiency | | Math Reasoning | | | | | OOD | | AVG |
| | | Tokens (M) | Speedup | AMC23 | GSM8K | MATH 500 | Minerva Math | Olympiad Bench | MMLU STEM | IFEval | |
|---|---|---|---|---|---|---|---|---|---|---|---|
| Base Model | 0 | - | - | 22.5 | 59.1 | 45.0 | 12.5 | 16.7 | 39.3 | 17.9 | 30.4 |
| GRPO | 10 | 65.8 | 1.00× | 37.5 | 74.0 | 54.0 | 18.8 | 22.8 | 43.9 | 20.3 | 38.8 |
| ↪ + Random Reuse | 10 | 58.0 | 1.11× | 37.5 | 74.1 | 57.8 | 21.0 | 21.0 | 42.9 | 22.0 | 39.5 |
| GRPO | 20 | 127.4 | 1.00× | 27.5 | 78.2 | 57.8 | 25.4 | 24.7 | 45.5 | 20.3 | 39.9 |
| ↪ + Random Reuse | 20 | 98.9 | 1.43× | 25.0 | 77.6 | 59.4 | 24.6 | 24.1 | 46.0 | 25.9 | 40.4 |
| GRPO | 30 | 187.7 | 1.00× | 37.5 | 80.1 | 60.4 | 22.4 | 25.0 | 47.9 | 20.9 | 42.0 |
| ↪ + Random Reuse | 30 | 134.1 | 1.61× | 35.0 | 78.9 | 63.2 | 26.8 | 24.9 | 50.6 | 29.2 | 44.1 |
| GRPO | 40 | 248.3 | 1.00× | 32.5 | 79.9 | 65.0 | 23.5 | 24.7 | 50.7 | 21.3 | 42.5 |
| ↪ + Random Reuse | 40 | 165.4 | 1.75× | 40.0 | 80.9 | 64.0 | 26.1 | 28.4 | 56.8 | 27.7 | 46.3 |
| GRPO | 50 | 309.1 | 1.00× | 35.0 | 81.2 | 64.2 | 25.7 | 24.6 | 53.7 | 25.1 | 44.2 |
| ↪ + Random Reuse | 50 | 194.1 | 1.89× | 35.0 | 81.0 | 63.2 | 27.9 | 27.0 | 57.7 | 25.7 | 45.4 |
| GRPO | 60 | 370.4 | 1.00× | 35.0 | 81.3 | 63.6 | 28.3 | 26.7 | 56.0 | 24.0 | 45.0 |
| ↪ + Random Reuse | 60 | 221.8 | 2.03× | 22.5 | 80.2 | 64.2 | 24.6 | 25.9 | 59.2 | 26.1 | 43.2 |
| GRPO | 70 | 431.9 | 1.00× | 42.5 | 82.3 | 61.6 | 26.5 | 28.1 | 55.5 | 26.2 | 46.1 |
| ↪ + Random Reuse | 70 | 249.6 | 2.14× | 32.5 | 80.7 | 63.2 | 27.2 | 26.1 | 60.9 | 25.5 | 45.2 |
| GRPO | 80 | 493.5 | 1.00× | 25.0 | 82.0 | 64.4 | 24.3 | 26.4 | 59.4 | 25.0 | 43.8 |
| ↪ + Random Reuse | 80 | 277.5 | 2.25× | 25.0 | 78.7 | 57.0 | 21.7 | 22.7 | 44.6 | 22.4 | 38.9 |
| GRPO | 90 | 554.8 | 1.00× | 42.5 | 82.6 | 64.4 | 26.5 | 25.5 | 60.7 | 24.4 | 46.7 |
| ↪ + Random Reuse | 90 | 304.5 | 2.35× | 37.5 | 80.0 | 60.4 | 21.7 | 25.3 | 53.1 | 24.0 | 43.1 |

Table 17: Intermediate training results of Qwen-3-1.7B-Base on DeepMath-6K with GRPO and SPEC-RL at lenience $\ell = 1$. We report rollout efficiency and accuracy every 10 training steps, illustrating the progression of model performance during training.

| Algorithm | Step | Rollout Efficiency | | Math Reasoning | | | | | OOD | | AVG |
| | | Tokens (M) | Speedup | AMC23 | GSM8K | MATH 500 | Minerva Math | Olympiad Bench | MMLU STEM | IFEval | |
|---|---|---|---|---|---|---|---|---|---|---|---|
| Base Model | 0 | - | - | 22.5 | 59.1 | 45.0 | 12.5 | 16.7 | 39.3 | 17.9 | 30.4 |
| GRPO | 10 | 65.8 | 1.00× | 37.5 | 74.0 | 54.0 | 18.8 | 22.8 | 43.9 | 20.3 | 38.8 |
| ↪ + SPEC-RL $\ell = 1$ | 10 | 63.5 | 1.03× | 30.0 | 74.8 | 55.2 | 20.2 | 21.2 | 44.5 | 18.9 | 37.8 |
| GRPO | 20 | 127.4 | 1.00× | 27.5 | 78.2 | 57.8 | 25.4 | 24.7 | 45.5 | 20.3 | 39.9 |
| ↪ + SPEC-RL $\ell = 1$ | 20 | 118.9 | 1.05× | 35.0 | 78.2 | 61.6 | 25.4 | 22.4 | 45.1 | 22.9 | 41.5 |
| GRPO | 30 | 187.7 | 1.00× | 37.5 | 80.1 | 60.4 | 22.4 | 25.0 | 47.9 | 20.9 | 42.0 |
| ↪ + SPEC-RL $\ell = 1$ | 30 | 168.0 | 1.07× | 20.0 | 79.5 | 61.2 | 25.0 | 25.0 | 49.9 | 23.7 | 40.6 |
| GRPO | 40 | 248.3 | 1.00× | 32.5 | 79.9 | 65.0 | 23.5 | 24.7 | 50.7 | 21.3 | 42.5 |
| ↪ + SPEC-RL $\ell = 1$ | 40 | 213.5 | 1.11× | 40.0 | 80.2 | 63.0 | 27.2 | 24.9 | 51.7 | 22.4 | 44.2 |
| GRPO | 50 | 309.1 | 1.00× | 35.0 | 81.2 | 64.2 | 25.7 | 24.6 | 53.7 | 25.1 | 44.2 |
| ↪ + SPEC-RL $\ell = 1$ | 50 | 257.0 | 1.14× | 42.5 | 80.3 | 63.2 | 29.0 | 24.7 | 54.0 | 24.4 | 45.4 |
| GRPO | 60 | 370.4 | 1.00× | 35.0 | 81.3 | 63.6 | 28.3 | 26.7 | 56.0 | 24.0 | 45.0 |
| ↪ + SPEC-RL $\ell = 1$ | 60 | 298.5 | 1.17× | 42.5 | 80.2 | 64.2 | 26.8 | 26.7 | 55.6 | 23.7 | 45.7 |
| GRPO | 70 | 431.9 | 1.00× | 42.5 | 82.3 | 61.8 | 26.5 | 28.1 | 55.5 | 26.2 | 46.1 |
| ↪ + SPEC-RL $\ell = 1$ | 70 | 339.2 | 1.19× | 35.0 | 81.0 | 62.6 | 28.7 | 28.0 | 58.0 | 26.8 | 45.7 |
| GRPO | 80 | 493.5 | 1.00× | 25.0 | 82.0 | 64.4 | 24.3 | 26.4 | 59.4 | 25.0 | 43.8 |
| ↪ + SPEC-RL $\ell = 1$ | 80 | 379.7 | 1.20× | 37.5 | 81.4 | 67.4 | 22.8 | 28.0 | 60.2 | 23.7 | 45.9 |
| GRPO | 90 | 554.8 | 1.00× | 42.5 | 82.6 | 64.4 | 26.5 | 25.5 | 60.7 | 24.4 | 46.7 |
| ↪ + SPEC-RL $\ell = 1$ | 90 | 419.1 | 1.22× | 40.0 | 81.8 | 63.8 | 28.7 | 26.5 | 59.6 | 25.9 | 46.6 |

Table 18: Intermediate training results of Qwen-3-1.7B-Base on DeepMath-6K with GRPO and SPEC-RL at lenience $\ell = e^{0.2}$. We report rollout efficiency and accuracy every 10 training steps, illustrating the progression of model performance during training.

| Algorithm | Step | Rollout Efficiency | | Math Reasoning | | | | | OOD | | AVG |
| | | Tokens (M) | Speedup | AMC23 | GSM8K | MATH 500 | Minerva Math | Olympiad Bench | MMLU STEM | IFEval | |
|---|---|---|---|---|---|---|---|---|---|---|---|
| Base Model | 0 | - | - | 22.5 | 59.1 | 45.0 | 12.5 | 16.7 | 39.3 | 17.9 | 30.4 |
| GRPO | 10 | 65.8 | 1.00× | 37.5 | 74.0 | 54.0 | 18.8 | 22.8 | 43.9 | 20.3 | 38.8 |
| ↪ + SPEC-RL $\ell = e^{0.2}$ | 10 | 53.6 | 1.19× | 32.5 | 73.2 | 55.6 | 19.1 | 22.1 | 42.4 | 20.0 | 37.8 |
| GRPO | 20 | 127.4 | 1.00× | 27.5 | 78.2 | 57.8 | 25.4 | 24.7 | 45.5 | 20.3 | 39.9 |
| ↪ + SPEC-RL $\ell = e^{0.2}$ | 20 | 92.7 | 1.29× | 40.0 | 78.8 | 60.6 | 25.4 | 25.5 | 46.2 | 20.0 | 42.5 |
| GRPO | 30 | 187.7 | 1.00× | 37.5 | 80.1 | 60.4 | 22.4 | 25.0 | 47.9 | 20.9 | 42.0 |
| ↪ + SPEC-RL $\ell = e^{0.2}$ | 30 | 120.5 | 1.42× | 32.5 | 79.6 | 60.8 | 26.8 | 27.0 | 49.5 | 22.9 | 42.7 |
| GRPO | 40 | 248.3 | 1.00× | 32.5 | 79.9 | 65.0 | 23.5 | 24.7 | 50.7 | 21.3 | 42.5 |
| ↪ + SPEC-RL $\ell = e^{0.2}$ | 40 | 143.6 | 1.54× | 32.5 | 80.3 | 61.8 | 28.3 | 26.4 | 51.8 | 22.2 | 43.3 |
| GRPO | 50 | 309.1 | 1.00× | 35.0 | 81.2 | 64.2 | 25.7 | 24.6 | 53.7 | 25.1 | 44.2 |
| ↪ + SPEC-RL $\ell = e^{0.2}$ | 50 | 166.1 | 1.62× | 37.5 | 80.4 | 65.2 | 27.6 | 25.9 | 54.4 | 25.5 | 45.2 |
| GRPO | 60 | 370.4 | 1.00× | 35.0 | 81.3 | 63.6 | 28.3 | 26.7 | 56.0 | 24.0 | 45.0 |
| ↪ + SPEC-RL $\ell = e^{0.2}$ | 60 | 186.1 | 1.70× | 37.5 | 81.7 | 63.6 | 29.4 | 25.5 | 55.0 | 24.4 | 45.3 |
| GRPO | 70 | 431.9 | 1.00× | 42.5 | 82.3 | 61.8 | 26.5 | 28.1 | 55.5 | 26.2 | 46.1 |
| ↪ + SPEC-RL $\ell = e^{0.2}$ | 70 | 206.5 | 1.77× | 42.5 | 80.4 | 64.6 | 27.2 | 29.5 | 58.4 | 25.1 | 46.8 |
| GRPO | 80 | 493.5 | 1.00× | 25.0 | 82.0 | 64.4 | 24.3 | 26.4 | 59.4 | 25.0 | 43.8 |
| ↪ + SPEC-RL $\ell = e^{0.2}$ | 80 | 226.3 | 1.82× | 32.5 | 81.8 | 63.4 | 29.4 | 29.9 | 57.9 | 24.0 | 45.6 |
| GRPO | 90 | 554.8 | 1.00× | 42.5 | 82.6 | 64.4 | 26.5 | 25.5 | 60.7 | 24.4 | 46.7 |
| ↪ + SPEC-RL $\ell = e^{0.2}$ | 90 | 246.7 | 1.86× | 37.5 | 83.3 | 66.4 | 29.8 | 29.6 | 58.5 | 25.9 | 47.3 |

Table 19: Intermediate training results of Qwen-3-1.7B-Base on DeepMath-6K with GRPO and SPEC-RL at lenience $\ell = e^{0.5}$. We report rollout efficiency and accuracy every 10 training steps, illustrating the progression of model performance during training.

| Algorithm | Step | Rollout Efficiency | | Math Reasoning | | | | | OOD | | AVG |
| | | Tokens (M) | Speedup | AMC23 | GSM8K | MATH 500 | Minerva Math | Olympiad Bench | MMLU STEM | IFEval | |
|---|---|---|---|---|---|---|---|---|---|---|---|
| Base Model | 0 | - | - | 22.5 | 59.1 | 45.0 | 12.5 | 16.7 | 39.3 | 17.9 | 30.4 |
| GRPO | 10 | 65.8 | 1.00× | 37.5 | 74.0 | 54.0 | 18.8 | 22.8 | 43.9 | 20.3 | 38.8 |
| ↪ + SPEC-RL $\ell = e^{0.5}$ | 10 | 43.6 | 1.41× | 27.5 | 75.7 | 55.8 | 21.7 | 21.8 | 43.0 | 22.4 | 38.3 |
| GRPO | 20 | 127.4 | 1.00× | 27.5 | 78.2 | 57.8 | 25.4 | 24.7 | 45.5 | 20.3 | 39.9 |
| ↪ + SPEC-RL $\ell = e^{0.5}$ | 20 | 67.2 | 1.66× | 30.0 | 80.0 | 63.0 | 25.0 | 24.6 | 46.9 | 22.2 | 41.7 |
| GRPO | 30 | 187.7 | 1.00× | 37.5 | 80.1 | 60.4 | 22.4 | 25.0 | 47.9 | 20.9 | 42.0 |
| ↪ + SPEC-RL $\ell = e^{0.5}$ | 30 | 85.1 | 1.85× | 30.0 | 81.0 | 64.0 | 25.4 | 27.6 | 51.1 | 26.2 | 43.6 |
| GRPO | 40 | 248.3 | 1.00× | 32.5 | 79.9 | 65.0 | 23.5 | 24.7 | 50.7 | 21.3 | 42.5 |
| ↪ + SPEC-RL $\ell = e^{0.5}$ | 40 | 102.9 | 1.96× | 37.5 | 80.7 | 63.8 | 26.5 | 26.1 | 52.2 | 23.5 | 44.3 |
| GRPO | 50 | 309.1 | 1.00× | 35.0 | 81.2 | 64.2 | 25.7 | 24.6 | 53.7 | 25.1 | 44.2 |
| ↪ + SPEC-RL $\ell = e^{0.5}$ | 50 | 119.4 | 2.06× | 32.5 | 81.1 | 64.4 | 28.7 | 28.0 | 55.6 | 27.7 | 45.4 |
| GRPO | 60 | 370.4 | 1.00× | 35.0 | 81.3 | 63.6 | 28.3 | 26.7 | 56.0 | 24.0 | 45.0 |
| ↪ + SPEC-RL $\ell = e^{0.5}$ | 60 | 135.1 | 2.14× | 27.5 | 83.4 | 66.4 | 26.5 | 29.9 | 54.7 | 28.8 | 45.3 |
| GRPO | 70 | 431.9 | 1.00× | 42.5 | 82.3 | 61.8 | 26.5 | 28.1 | 55.5 | 26.2 | 46.1 |
| ↪ + SPEC-RL $\ell = e^{0.5}$ | 70 | 153.2 | 2.18× | 40.0 | 82.5 | 65.4 | 26.5 | 29.9 | 55.6 | 27.4 | 46.8 |
| GRPO | 80 | 493.5 | 1.00× | 25.0 | 82.0 | 64.4 | 24.3 | 26.4 | 59.4 | 25.0 | 43.8 |
| ↪ + SPEC-RL $\ell = e^{0.5}$ | 80 | 168.1 | 2.24× | 45.0 | 83.7 | 67.0 | 29.8 | 29.6 | 57.1 | 28.3 | 48.6 |
| GRPO | 90 | 554.8 | 1.00× | 42.5 | 82.6 | 64.4 | 26.5 | 25.5 | 60.7 | 24.4 | 46.7 |
| ↪ + SPEC-RL $\ell = e^{0.5}$ | 90 | 182.7 | 2.29× | 37.5 | 84.4 | 68.0 | 29.4 | 29.3 | 58.3 | 28.8 | 48.0 |

Table 20: Intermediate training results of Qwen-3-1.7B-Base on DeepMath-6K with GRPO and SPEC-RL at lenience $\ell = e^{0.8}$. We report rollout efficiency and accuracy every 10 training steps, illustrating the progression of model performance during training.

| Algorithm | Step | Rollout Efficiency | | Math Reasoning | | | | | OOD | | AVG |
| | | Tokens (M) | Speedup | AMC23 | GSM8K | MATH 500 | Minerva Math | Olympiad Bench | MMLU STEM | IFEval | |
|---|---|---|---|---|---|---|---|---|---|---|---|
| Base Model | 0 | - | - | 22.5 | 59.1 | 45.0 | 12.5 | 16.7 | 39.3 | 17.9 | 30.4 |
| GRPO | 10 | 65.8 | 1.00× | 37.5 | 74.0 | 54.0 | 18.8 | 22.8 | 43.9 | 20.3 | 38.8 |
| ↪ + SPEC-RL $\ell = e^{0.8}$ | 10 | 41.9 | 1.46× | 30.0 | 76.6 | 57.0 | 19.9 | 21.6 | 44.8 | 20.1 | 38.6 |
| GRPO | 20 | 127.4 | 1.00× | 27.5 | 78.2 | 57.8 | 25.4 | 24.7 | 45.5 | 20.3 | 39.9 |
| ↪ + SPEC-RL $\ell = e^{0.8}$ | 20 | 57.5 | 1.86× | 27.5 | 80.0 | 60.0 | 26.1 | 24.4 | 46.2 | 23.3 | 41.1 |
| GRPO | 30 | 187.7 | 1.00× | 37.5 | 80.1 | 60.4 | 22.4 | 25.0 | 47.9 | 20.9 | 42.0 |
| ↪ + SPEC-RL $\ell = e^{0.8}$ | 30 | 70.1 | 2.12× | 37.5 | 81.6 | 62.4 | 28.7 | 24.0 | 51.5 | 27.4 | 44.7 |
| GRPO | 40 | 248.3 | 1.00× | 32.5 | 79.9 | 65.0 | 23.5 | 24.7 | 50.7 | 21.3 | 42.5 |
| ↪ + SPEC-RL $\ell = e^{0.8}$ | 40 | 81.8 | 2.29× | 37.5 | 82.0 | 63.8 | 26.8 | 27.7 | 53.8 | 28.5 | 45.7 |
| GRPO | 50 | 309.1 | 1.00× | 35.0 | 81.2 | 64.2 | 25.7 | 24.6 | 53.7 | 25.1 | 44.2 |
| ↪ + SPEC-RL $\ell = e^{0.8}$ | 50 | 97.5 | 2.35× | 35.0 | 81.5 | 63.4 | 28.3 | 26.5 | 57.0 | 28.7 | 45.8 |
| GRPO | 60 | 370.4 | 1.00× | 35.0 | 81.3 | 63.6 | 28.3 | 26.7 | 56.0 | 24.0 | 45.0 |
| ↪ + SPEC-RL $\ell = e^{0.8}$ | 60 | 110.8 | 2.43× | 47.5 | 82.2 | 61.8 | 26.8 | 25.8 | 57.7 | 25.5 | 46.8 |
| GRPO | 70 | 431.9 | 1.00× | 42.5 | 82.3 | 61.8 | 26.5 | 28.1 | 55.5 | 26.2 | 46.1 |
| ↪ + SPEC-RL $\ell = e^{0.8}$ | 70 | 120.0 | 2.54× | 35.0 | 84.2 | 62.2 | 26.8 | 25.9 | 58.9 | 27.9 | 45.8 |
| GRPO | 80 | 493.5 | 1.00× | 25.0 | 82.0 | 64.4 | 24.3 | 26.4 | 59.4 | 25.0 | 43.8 |
| ↪ + SPEC-RL $\ell = e^{0.8}$ | 80 | 132.0 | 2.60× | 32.5 | 84.1 | 63.6 | 27.6 | 26.5 | 58.4 | 28.3 | 45.9 |
| GRPO | 90 | 554.8 | 1.00× | 42.5 | 82.6 | 64.4 | 26.5 | 25.5 | 60.7 | 24.4 | 46.7 |
| ↪ + SPEC-RL $\ell = e^{0.8}$ | 90 | 144.8 | 2.64× | 37.5 | 83.5 | 63.6 | 27.2 | 25.0 | 61.7 | 26.2 | 46.4 |

Table 21: Intermediate training results of Qwen-3-1.7B-Base on DeepMath-6K with GRPO and SPEC-RL at lenience $\ell = e^{1.0}$. We report rollout efficiency and accuracy every 10 training steps, illustrating the progression of model performance during training.

| Algorithm | Step | Rollout Efficiency | | Math Reasoning | | | | | OOD | | AVG |
| | | Tokens (M) | Speedup | AMC23 | GSM8K | MATH 500 | Minerva Math | Olympiad Bench | MMLU STEM | IFEval | |
|---|---|---|---|---|---|---|---|---|---|---|---|
| Base Model | 0 | - | - | 22.5 | 59.1 | 45.0 | 12.5 | 16.7 | 39.3 | 17.9 | 30.4 |
| GRPO | 10 | 65.8 | 1.00× | 37.5 | 74.0 | 54.0 | 18.8 | 22.8 | 43.9 | 20.3 | 38.8 |
| ↪ + SPEC-RL $\ell = e^{1.0}$ | 10 | 41.5 | 1.46× | 25.0 | 75.7 | 55.0 | 17.3 | 23.3 | 44.2 | 18.5 | 37.0 |
| GRPO | 20 | 127.4 | 1.00× | 27.5 | 78.2 | 57.8 | 25.4 | 24.7 | 45.5 | 20.3 | 39.9 |
| ↪ + SPEC-RL $\ell = e^{1.0}$ | 20 | 52.6 | 1.99× | 25.0 | 79.0 | 60.0 | 25.0 | 22.4 | 46.7 | 23.1 | 40.2 |
| GRPO | 30 | 187.7 | 1.00× | 37.5 | 80.1 | 60.4 | 22.4 | 25.0 | 47.9 | 20.9 | 42.0 |
| ↪ + SPEC-RL $\ell = e^{1.0}$ | 30 | 64.8 | 2.23× | 42.5 | 80.8 | 64.4 | 25.4 | 27.4 | 49.9 | 30.1 | 45.8 |
| GRPO | 40 | 248.3 | 1.00× | 32.5 | 79.9 | 65.0 | 23.5 | 24.7 | 50.7 | 21.3 | 42.5 |
| ↪ + SPEC-RL $\ell = e^{1.0}$ | 40 | 73.2 | 2.46× | 32.5 | 81.3 | 65.0 | 24.6 | 27.7 | 52.1 | 27.2 | 44.3 |
| GRPO | 50 | 309.1 | 1.00× | 35.0 | 81.2 | 64.2 | 25.7 | 24.6 | 53.7 | 25.1 | 44.2 |
| ↪ + SPEC-RL $\ell = e^{1.0}$ | 50 | 86.6 | 2.55× | 30.0 | 83.2 | 62.0 | 25.0 | 25.9 | 53.4 | 28.7 | 44.0 |
| GRPO | 60 | 370.4 | 1.00× | 35.0 | 81.3 | 63.6 | 28.3 | 26.7 | 56.0 | 24.0 | 45.0 |
| ↪ + SPEC-RL $\ell = e^{1.0}$ | 60 | 97.8 | 2.64× | 32.5 | 84.1 | 62.0 | 28.3 | 27.1 | 52.9 | 28.1 | 45.0 |
| GRPO | 70 | 431.9 | 1.00× | 42.5 | 82.3 | 61.8 | 26.5 | 28.1 | 55.5 | 26.2 | 46.1 |
| ↪ + SPEC-RL $\ell = e^{1.0}$ | 70 | 107.7 | 2.72× | 35.0 | 83.0 | 62.8 | 25.4 | 27.0 | 52.1 | 27.9 | 44.7 |
| GRPO | 80 | 493.5 | 1.00× | 25.0 | 82.0 | 64.4 | 24.3 | 26.4 | 59.4 | 25.0 | 43.8 |
| ↪ + SPEC-RL $\ell = e^{1.0}$ | 80 | 116.6 | 2.81× | 30.0 | 82.3 | 63.4 | 26.1 | 27.0 | 54.0 | 29.0 | 44.5 |
| GRPO | 90 | 554.8 | 1.00× | 42.5 | 82.6 | 64.4 | 26.5 | 25.5 | 60.7 | 24.4 | 46.7 |
| ↪ + SPEC-RL $\ell = e^{1.0}$ | 90 | 123.0 | 2.91× | 37.5 | 83.9 | 62.4 | 25.7 | 24.9 | 54.8 | 28.3 | 45.4 |

Table 22: Intermediate training results of Qwen-3-1.7B-Base on DeepMath-6K with GRPO and SPEC-RL at lenience $\ell = e^{2.0}$. We report rollout efficiency and accuracy every 10 training steps, illustrating the progression of model performance during training.

| Algorithm | Step | Rollout Efficiency | | Math Reasoning | | | | | OOD | | AVG |
| | | Tokens (M) | Speedup | AMC23 | GSM8K | MATH 500 | Minerva Math | Olympiad Bench | MMLU STEM | IFEval | |
|---|---|---|---|---|---|---|---|---|---|---|---|
| Base Model | 0 | - | - | 22.5 | 59.1 | 45.0 | 12.5 | 16.7 | 39.3 | 17.9 | 30.4 |
| GRPO | 10 | 65.8 | 1.00× | 37.5 | 74.0 | 54.0 | 18.8 | 22.8 | 43.9 | 20.3 | 38.8 |
| ↪ + SPEC-RL $\ell = e^{2.0}$ | 10 | 40.4 | 1.50× | 37.5 | 75.3 | 55.4 | 21.7 | 21.8 | 44.0 | 19.6 | 39.3 |
| GRPO | 20 | 127.4 | 1.00× | 27.5 | 78.2 | 57.8 | 25.4 | 24.7 | 45.5 | 20.3 | 39.9 |
| ↪ + SPEC-RL $\ell = e^{2.0}$ | 20 | 43.9 | 2.24× | 25.0 | 78.9 | 60.4 | 28.3 | 23.7 | 44.3 | 25.0 | 40.8 |
| GRPO | 30 | 187.7 | 1.00× | 37.5 | 80.1 | 60.4 | 22.4 | 25.0 | 47.9 | 20.9 | 42.0 |
| ↪ + SPEC-RL $\ell = e^{2.0}$ | 30 | 51.2 | 2.59× | 30.0 | 81.1 | 61.0 | 26.1 | 25.6 | 49.8 | 29.2 | 43.3 |
| GRPO | 40 | 248.3 | 1.00× | 32.5 | 79.9 | 65.0 | 23.5 | 24.7 | 50.7 | 21.3 | 42.5 |
| ↪ + SPEC-RL $\ell = e^{2.0}$ | 40 | 54.5 | 2.93× | 25.0 | 80.7 | 64.0 | 28.7 | 26.4 | 52.5 | 31.4 | 44.1 |
| GRPO | 50 | 309.1 | 1.00× | 35.0 | 81.2 | 64.2 | 25.7 | 24.6 | 53.7 | 25.1 | 44.2 |
| ↪ + SPEC-RL $\ell = e^{2.0}$ | 50 | 60.1 | 3.14× | 37.5 | 82.1 | 61.6 | 25.7 | 24.1 | 52.4 | 29.2 | 44.7 |
| GRPO | 60 | 370.4 | 1.00× | 35.0 | 81.3 | 63.6 | 28.3 | 26.7 | 56.0 | 24.0 | 45.0 |
| ↪ + SPEC-RL $\ell = e^{2.0}$ | 60 | 67.2 | 3.24× | 32.5 | 83.5 | 57.8 | 25.0 | 24.3 | 54.6 | 31.4 | 44.2 |
| GRPO | 70 | 431.9 | 1.00× | 42.5 | 82.3 | 61.8 | 26.5 | 28.1 | 55.5 | 26.2 | 46.1 |
| ↪ + SPEC-RL $\ell = e^{2.0}$ | 70 | 76.8 | 3.28× | 22.5 | 82.4 | 57.4 | 21.7 | 23.7 | 52.5 | 28.7 | 41.3 |
| GRPO | 80 | 493.5 | 1.00× | 25.0 | 82.0 | 64.4 | 24.3 | 26.4 | 59.4 | 25.0 | 43.8 |
| ↪ + SPEC-RL $\ell = e^{2.0}$ | 80 | 90.5 | 3.24× | 27.5 | 81.3 | 57.4 | 19.5 | 23.4 | 54.8 | 30.3 | 42.0 |
| GRPO | 90 | 554.8 | 1.00× | 42.5 | 82.6 | 64.4 | 26.5 | 25.5 | 60.7 | 24.4 | 46.7 |
| ↪ + SPEC-RL $\ell = e^{2.0}$ | 90 | 114.4 | 3.05× | 30.0 | 80.4 | 55.0 | 21.0 | 21.9 | 53.5 | 29.0 | 41.5 |

Table 23: Intermediate training results of Qwen-3-1.7B-Base on DeepMath-6K with GRPO and SPEC-RL at lenience $\ell = \infty$. We report rollout efficiency and accuracy every 10 training steps, illustrating the progression of model performance during training.

| Algorithm | Step | Rollout Efficiency | | Math Reasoning | | | | | OOD | | AVG |
| | | Tokens (M) | Speedup | AMC23 | GSM8K | MATH 500 | Minerva Math | Olympiad Bench | MMLU STEM | IFEval | |
|---|---|---|---|---|---|---|---|---|---|---|---|
| Base Model | 0 | - | - | 22.5 | 59.1 | 45.0 | 12.5 | 16.7 | 39.3 | 17.9 | 30.4 |
| GRPO | 10 | 65.8 | 1.00× | 37.5 | 74.0 | 54.0 | 18.8 | 22.8 | 43.9 | 20.3 | 38.8 |
| ↪ + SPEC-RL $\ell = \infty$ | 10 | 40.0 | 1.75× | 17.5 | 71.7 | 53.8 | 16.9 | 21.6 | 42.5 | 20.0 | 34.9 |
| GRPO | 20 | 127.4 | 1.00× | 27.5 | 78.2 | 57.8 | 25.4 | 24.7 | 45.5 | 20.3 | 39.9 |
| ↪ + SPEC-RL $\ell = \infty$ | 20 | 40.0 | 3.39× | 35.0 | 78.0 | 56.4 | 21.0 | 22.7 | 44.4 | 21.4 | 39.8 |
| GRPO | 30 | 187.7 | 1.00× | 37.5 | 80.1 | 60.4 | 22.4 | 25.0 | 47.9 | 20.9 | 42.0 |
| ↪ + SPEC-RL $\ell = \infty$ | 30 | 40.0 | 5.03× | 35.0 | 76.9 | 53.2 | 21.7 | 21.3 | 43.0 | 21.4 | 38.9 |
| GRPO | 40 | 248.3 | 1.00× | 32.5 | 79.9 | 65.0 | 23.5 | 24.7 | 50.7 | 21.3 | 42.5 |
| ↪ + SPEC-RL $\ell = \infty$ | 40 | 40.0 | 6.65× | 32.5 | 76.7 | 57.0 | 17.3 | 22.4 | 43.3 | 22.6 | 38.8 |
| GRPO | 50 | 309.1 | 1.00× | 35.0 | 81.2 | 64.2 | 25.7 | 24.6 | 53.7 | 25.1 | 44.2 |
| ↪ + SPEC-RL $\ell = \infty$ | 50 | 40.0 | 8.29× | 37.5 | 77.4 | 57.0 | 23.2 | 22.5 | 43.3 | 22.7 | 40.5 |
| GRPO | 60 | 370.4 | 1.00× | 35.0 | 81.3 | 63.6 | 28.3 | 26.7 | 56.0 | 24.0 | 45.0 |
| ↪ + SPEC-RL $\ell = \infty$ | 60 | 40.0 | 9.93× | 35.0 | 77.1 | 58.8 | 23.5 | 22.8 | 42.6 | 23.7 | 40.5 |
| GRPO | 70 | 431.9 | 1.00× | 42.5 | 82.3 | 61.8 | 26.5 | 28.1 | 55.5 | 26.2 | 46.1 |
| ↪ + SPEC-RL $\ell = \infty$ | 70 | 40.0 | 11.56× | 35.0 | 77.9 | 58.6 | 23.2 | 22.4 | 44.0 | 20.9 | 40.3 |
| GRPO | 80 | 493.5 | 1.00× | 25.0 | 82.0 | 64.4 | 24.3 | 26.4 | 59.4 | 25.0 | 43.8 |
| ↪ + SPEC-RL $\ell = \infty$ | 80 | 40.0 | 13.21× | 32.5 | 79.0 | 60.2 | 23.9 | 24.3 | 42.8 | 23.3 | 40.9 |
| GRPO | 90 | 554.8 | 1.00× | 42.5 | 82.6 | 64.4 | 26.5 | 25.5 | 60.7 | 24.4 | 46.7 |
| ↪ + SPEC-RL $\ell = \infty$ | 90 | 40.0 | 14.86× | 32.5 | 78.1 | 60.4 | 19.9 | 23.7 | 44.1 | 22.0 | 40.1 |

# B  CASE STUDY

To provide a more intuitive understanding of how SPEC-RL operates during training, we present several case studies comparing cached rollouts from previous epochs with newly generated rollouts under the current policy. These examples highlight how speculative prefixes are verified and reused, and how continuation is triggered once a rejection occurs. They also illustrate typical scenarios where SPEC-RL improves efficiency by avoiding redundant generation, while still correcting erroneous reasoning steps when necessary. Representative cases are shown in Figures 13, 14, 15, and 16.

---

**Prompt**

**system**
You are a helpful assistant.
**user**
Kim's birthday was 200 days ago. Today is Wednesday. On what day of the week did his birthday fall?
Please reason step by step, and put your final answer within \boxed{}.
**assistant**

---

**Old Rollout from Epoch$_{t-1}$**

To determine the day of the week Kim's birthday fell on, we can follow these steps:

1. Identify the current day and the number of days since Kim's birthday.
   - Today is Wednesday.
   - Kim's birthday was 200 days ago.

2. Determine the day of the week 200 days ago from Wednesday.
   - We need to find the remainder when 200 is divided by 7, because there are 7 days in a week. This will tell us how many days forward or backward from Wednesday we need to count.
   - Calculate $200 \div 7$:
$$200 \div 7 = 28 \text{ remainder } 4$$
   This means that 200 days is equivalent to 28 weeks and 4 days. So, 200 days ago from Wednesday is the same as 4 days before Wednesday.

3. Count backward 4 days from Wednesday.
   - Monday is 1 day before Wednesday.
   - Sunday is 2 days before Wednesday.
   - Saturday is 3 days before Wednesday.
   - Friday is 4 days before Wednesday.

Therefore, 200 days ago from Wednesday was a Friday. 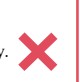
The final answer is \boxed{Friday}.

**New Rollout from Epoch$_t$**

To determine the day of the week Kim's birthday fell on, we can follow these steps:

1. Identify the current day and the number of days since Kim's birthday.
   - Today is Wednesday.
   - Kim's birthday was 200 days ago.

2. Determine the day of the week 200 days ago from Wednesday.
   - We need to find the remainder when 200 is divided by 7, because there are 7 days in a week. This will tell us how many days forward or backward from Wednesday we need to count.
   - Calculate the remainder:
$$200 \div 7 = 28 \text{ remainder } 4$$
   - This means that 200 days ago was 4 days before Wednesday.

3. Count backwards from Wednesday by 4 days.
   - Wednesday - 1 day = Tuesday
   - Tuesday - 1 day = Monday
   - Monday - 1 day = Sunday
   - Sunday - 1 day = Saturday

Therefore, Kim's birthday fell on a \boxed{\text{Saturday}}. 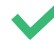

Figure 13: Case study comparing rollouts from previous and current training steps. The prompt denotes the model input. The old rollout and new rollout are generated by the respective model from corresponding epochs. Tokens highlighted in yellow indicate the verified speculative prefix. The red box marks incorrect reasoning steps, whereas the green box highlights correct reasoning steps.

**Prompt**

**system**
You are a helpful assistant.
**user**
The arithmetic mean of these five expressions is 24. What is the value of $x$?
$$x + 8 \qquad 15 \qquad 2x \qquad 13 \qquad 2x + 4$$
Please reason step by step, and put your final answer within \boxed{}.
**assistant**

**Old Rollout from Epoch$_{t-1}$**

To find the value of $x$, we start by understanding the arithmetic mean of the given expressions and setting it equal to 24. The expressions are:
1. $x + 8$
2. 15
3. $2x$
4. 13
5. $2x + 4$

The arithmetic mean of these expressions is given by:
$$\frac{(x + 8) + 15 + 2x + 13 + (2x + 4)}{5} = 24$$

First, we simplify and sum the expressions in the numerator:
$$\frac{(x + 8) + 15 + 2x + 13 + (2x + 4)}{5}$$
$$= \frac{x + 8 + 15 + 2x + 13 + 2x + 4}{5}$$
$$= \frac{5x + 30}{5}$$

Setting this equal to 24, we get:
$$\frac{5x + 30}{5} = 24$$

Next, we eliminate the fraction by multiplying both sides of the equation by 5:
$$5x + 30 = 120$$

Then, we isolate $x$ by subtracting 30 from both sides:
$$5x = 90$$

Finally, we solve for $x$ by dividing both sides by 5:
$$x = \frac{90}{5} = 18 \qquad \text{✗}$$

So, the value of $x$ is \boxed{18}.

**New Rollout from Epoch$_{t}$**

To find the value of $x$, we start by understanding the arithmetic mean of the given expressions and setting it equal to 24. The expressions are:
1. $x + 8$
2. 15
3. $2x$
4. 13
5. $2x + 4$

The arithmetic mean of these expressions is given by:
$$\frac{(x + 8) + 15 + 2x + 13 + (2x + 4)}{5} = 24$$

First, we simplify the numerator:
$$(x + 8) + 15 + 2x + 13 + (2x + 4)$$
$$= x + 8 + 15 + 2x + 13 + 2x + 4$$
$$= 5x + 40$$

So the equation becomes:
$$\frac{5x + 40}{5} = 24$$

Next, we eliminate the denominator by multiplying both sides of the equation by 5:
$$5x + 40 = 120$$

Then, we solve for $x$ by isolating it on one side of the equation. First, subtract 40 from both sides:
$$5x = 80$$

Finally, divide both sides by 5:
$$x = 16$$

Thus, the value of $x$ is \boxed{16}. ✓

Figure 14: Case study comparing rollouts from previous and current training steps. The prompt denotes the model input. The old rollout and new rollout are generated by the respective model from corresponding epochs. Tokens highlighted in yellow indicate the verified prefix. The red box marks incorrect reasoning steps, whereas the green box highlights correct reasoning steps.

**Prompt**

**system**
You are a helpful assistant.
**user**
Find the units digit of $9^{8^7}$.
Please reason step by step, and put your final answer within \boxed{}.
**assistant**

---

**Old Rollout from Epoch$_{t-1}$**

To find the units digit of $9^{8^7}$, we need to observe the pattern in the units digit of powers of 9.

Let's start by computing the first few powers of 9 and noting their units digits:
$$9^1 = 9 \ (\textit{units digit is } 9)$$
$$9^2 = 81 \ (\textit{units digit is } 1)$$
$$9^3 = 729 \ (\textit{units digit is } 9)$$
$$9^4 = 6561 \ (\textit{units digit is } 1)$$

From the above calculations, we see that the units digits repeat every 2 powers: 9, 1, 9, 1, ...

So, the units digit of $9^n$ is 9 if n is odd, and 1 if n is even.

Now, we need to determine if $8^7$ is odd or even.

Since $8^7$ is an odd power of 8, it will be odd as well (because any positive power of an even number is even).

Therefore, the units digit of $9^{8^7}$ will be the same as the units digit of $9^1$, which is 9.

Thus, the units digit of $9^{8^7}$ is **\boxed{9}**. ✗

---

**New Rollout from Epoch$_t$**

To find the units digit of $9^{8^7}$, we need to observe the pattern in the units digit of powers of 9.

Let's start by computing the first few powers of 9 and noting their units digits:
$$9^1 = 9 \ (\textit{units digit is } 9)$$
$$9^2 = 81 \ (\textit{units digit is } 1)$$
$$9^3 = 729 \ (\textit{units digit is } 9)$$
$$9^4 = 6561 \ (\textit{units digit is } 1)$$

From the above, we can see that the units digits repeat every 2 powers: 9, 1, 9, 1, ...

So, the units digit of $9^n$ is 9 if n is odd, and 1 if n is even.

Next, we need to determine if $8^7$ is odd or even. Since 8 is even, any power of 8 will also be even. Therefore, $8^7$ is an even number.

Since $8^7$ is even, the units digit of $9^{8^7}$ will be 1.

Thus, the units digit of $9^{8^7}$ is **\boxed{1}**. ✓

---

Figure 15: Case study comparing rollouts from previous and current training steps. The prompt denotes the model input. The old rollout and new rollout are generated by the respective model from corresponding epochs. Tokens highlighted in yellow indicate the verified speculative prefix. The red box marks incorrect reasoning steps, whereas the green box highlights correct reasoning steps.

**Prompt**

**system**
You are a helpful assistant.
**user**
If $-3 \leq x + 5 \leq 8$ and x is an integer, what is the sum of all the possible solutions?
Please reason step by step, and put your final answer within \boxed{}.
**assistant**

---

**Old Rollout from Epoch$_{t-1}$**

To solve the given inequality $-3 \leq x + 5 \leq 8$, we need to isolate x on one side of the inequality. We can do this by subtracting 5 from all parts of the inequality:
$$-3 - 5 \leq x + 5 - 5 \leq 8 - 5$$

Simplifying each part:
$$-8 \leq x \leq 3$$

Since x is an integer, the possible values for x are:
$$\{-8, -7, -6, -5, -4, -3, -2, -1, 0, 1, 2, 3\}$$

Next, we sum these possible values:
$$-8 + (-7) + (-6) + (-5) + (-4) + (-3) + (-2) + (-1)$$
$$+ 0 + 1 + 2 + 3$$

We can group the positive and negative terms:
$$(-8 + 3) + (-7 + 2) + (-6 + 1) + (-5 + 4) - 10$$
$$= -5 - 5 - 5 - 10 = -25$$

Thus, the sum of all the possible solutions is:
**\boxed{-25}.**                                    ✗

---

**New Rollout from Epoch$_t$**

To solve the given inequality $-3 \leq x + 5 \leq 8$, we need to isolate x on one side of the inequality. We can do this by subtracting 5 from all parts of the inequality:
$$-3 - 5 \leq x + 5 - 5 \leq 8 - 5$$

Simplifying each part:
$$-8 \leq x \leq 3$$

Since x is an integer, the possible values for x are:
$$\{-8, -7, -6, -5, -4, -3, -2, -1, 0, 1, 2, 3\}$$

Next, we need to find the sum of all these possible values of x. We can sum them directly:
$$-8 + (-7) + (-6) + (-5) + (-4) + (-3) + (-2) + (-1)$$
$$+ 0 + 1 + 2 + 3$$

We can simplify this by grouping the terms to make the addition easier:
$$(-8 + 3) + (-7 + 2) + (-6 + 1) + (-5 + 0)$$
$$+ (-4 + (-2)) + (-3 + (-1))$$
$$= -5 + (-5) + (-5) + (-5) + (-6) + (-4) = -30$$

Thus, the sum of all the possible solutions is:
**\boxed{-30}.**                                    ✓

Figure 16: Case study comparing rollouts from previous and current training steps. The prompt denotes the model input. The old rollout and new rollout are generated by the respective model from corresponding epochs. Tokens highlighted in yellow indicate the verified speculative prefix. The red box marks incorrect reasoning steps, whereas the green box highlights correct reasoning steps.