# OpenReview forum: "SPEC-RL: Accelerating On-Policy Reinforcement Learning via Speculative Rollouts"
_ICLR.cc/2026/Conference — ICLR 2026 Conference Withdrawn Submission_

### Official Review · Reviewer_HGQB · 2025-10-29

**Soundness:** 3
**Presentation:** 3
**Contribution:** 3
**Rating:** 4
**Confidence:** 3

**Summary:**

This paper proposes Spec-RL, a framework that leverages speculative sampling to accelerate the rollout phase in LLM-based reinforcement learning. Spec-RL stores trajectories from all rollouts. When the same prompt reappears, the response from the previous epoch is retrieved and tested to determine the reusable prefix. A lenience parameter is introduced to balance efficiency and accuracy. Experiments demonstrate 1.94× to 2.88× speedups in the rollout phase.

**Strengths:**

- The proposed method is both novel and practical. Since rollout is the major efficiency bottleneck in LLM-based reinforcement learning, the idea of accelerating it by reusing previous rollout prefixes is promising. The mechanism is well motivated through speculative sampling, and the method design is clear and easy to follow.

- Experiments are conducted on three different models, demonstrating the generality of the approach. The method achieves around 2× rollout speedup, which is strong. The paper also provides a detailed discussion on how the lenience parameter is tuned.

**Weaknesses:**

1. One unclear aspect is how the loss and importance sampling are handled. Suppose the reused prefix is o1 and the newly generated continuation is o2. Is the policy loss applied to o1 + o2 or only to o2? In addition, since the reused prefix has a log probability under the old model and two new log probabilities under the current policy, how are these values combined to form the importance sampling ratio?

2. The paper only considers reusing responses from the previous epoch. Would it be beneficial to also reuse responses from older epochs to further improve efficiency? Even if it brings limited gain, a discussion on this topic would be valuable.

**Questions:**

Please refer to the weakness part, and a following question:

- In my understanding, number of epochs in RL-for-LLM post training is typically small, and in some other cases, the data is generated during interaction with environment, which is hardly repeating. Thus I have a question that whether the proposed method is only workable in some settings while hardly workable in other. Could the authors have some discussion about their understanding of this concern?

---

> ### Author Response · Authors · 2025-11-19
> **Reponse to Reviewer HGQB**
>
> **(1) Clarification on loss and importance sampling**
>
> > *“Is the policy loss applied to o₁ + o₂ or only to o₂ … how are log probs combined?”*
>
> Implementation-wise, the reused prefix (`o₁`) and the newly generated continuation (`o₂`) are concatenated, and the **verl** framework recomputes log probs for the entire trajectory under the current policy.
> Thus, the loss is applied to the *whole sequence* as usual, and SPEC-RL modifies only the rollout phase.
>
> **(2) On reusing older epochs**
>
> > *“Would it be beneficial to also reuse responses from older epochs?”*
>
> Intuitively, reusing responses from earlier epochs may decrease efficacy because the distribution gap widens as the current policy diverges further from the old one. To validate this, we implemented a variant that reuses rollouts from two epochs prior. The results show that this leads to noticeably shorter accepted prefixes, which reduces the acceleration to around 1.26–1.44×, compared to the 1.4–2.3× achieved by standard SPEC-RL. Accuracy also falls slightly below the standard version. We also updated our evaluation to more stable metrics—AMC23-AVG@16 and AIME24/25-AVG@32—to better reflect the model’s reasoning ability across configurations.
>
> | Algorithm                 | Tokens(M) | Speedup | AMC23&#x0A;AVG@16 | AIME24&#x0A;AVG@32 | AIME25&#x0A;AVG@32 | GSM8K | MATH500 | Minerva | Olympiad | MMLU-STEM | IFEval | AVG  |
> | ------------------------- | --------- | ------- | ------------ | ------------- | ------------- | ----- | ------- | ------- | -------- | --------- | ------ | ---- |
> | -                         | -         | -       | 21.9         | 2.5           | 2.5           | 59.1  | 45      | 12.5    | 16.7     | 39.3      | 17.9   | 24.2 |
> | GRPO                      | 554.8     | 1.00    | 35.0         | 7.8           | 4.2           | 82.6  | 64.4    | 26.5    | 25.5     | 60.7      | 24.4   | 36.8 |
> | with SPEC-RL              | 182.7     | 2.29    | 38.3         | 9.1           | 4.9           | 84.4  | 68.0    | 29.4    | 29.3     | 58.3      | 28.8   | 38.9 |
> | with SPEC-RL-older-policy | 308.8     | 1.44    | 35.6         | 9.0           | 4.7           | 83.8  | 66.4    | 29.0    | 28.6     | 57.3      | 28.3   | 38.1 |
>
> These results collectively indicate that speculative reuse in RLVR benefits from the **distributional proximity of adjacent epochs**, and that even a small increase in the age of the reused policy already weakens this proximity. Therefore, using the **immediately previous epoch** provides the most effective balance between reuse quality, stability, and acceleration, which is why it is adopted as the default setting in SPEC-RL.
>
> **(3) Applicability across training regimes**
>
> > *“Whether the proposed method is only workable in some settings while hardly workable in others?”*
>
> As far as we know, RL post-training practice typically uses a relatively small but high-quality dataset together with a large batch size to stabilize optimization. This setup naturally produces **many training epochs**, often from dozens to well over a hundred. For example, **DAPO** runs for around 150 epochs [1], and the **Qwen3** technical report shows that their configuration (≈3.9k samples with batch sizes of 256–2048) results in 10–85 epochs [2]. These are representative regimes in which repeated epochs over the same prompt set are intrinsic to the training process, making them **exactly the scenarios where SPEC-RL provides substantial acceleration**, since adjacent epochs revisit highly similar rollouts.
>
> Regarding the agent’s environment, most current RL-for-LLM research still operates in **nearly static environments** (e.g., Python interpreters or tool-augmented knowledge systems), where trajectory distributions remain stable across epochs and SPEC-RL naturally applies. RL in dynamic, rapidly changing environments is indeed an interesting open direction, and we will incorporate discussion on such settings in the revised version.
>
>
> [1] DAPO: An Open-Source LLM Reinforcement Learning System at Scale. https://arxiv.org/pdf/2503.14476
>
> [2] Qwen3 Technical Report. https://arxiv.org/abs/2505.09388

---

### Official Review · Reviewer_8qjz · 2025-11-01

**Soundness:** 3
**Presentation:** 3
**Contribution:** 3
**Rating:** 2
**Confidence:** 5

**Summary:**

This paper introduces SPEC-RL, a method that incorporates speculative decoding into the rollout stage of RL training to improve efficiency. The approach reuses previously generated segments as prefixes, thereby reducing redundant computation and achieving substantial speedup according to the reported experiments.

**Strengths:**

See weaknesses.

**Weaknesses:**

The submission does not comply with the ICLR formatting requirements. The page layout deviates from the official ICLR template. Specifically, the side margins are significantly smaller than in the standard format. As a result, the actual body text occupies more space per page, effectively exceeding the permitted page limit. This makes the submission non-compliant with the conference’s formatting and length guidelines.

Given the extent of the formatting irregularity and the fact that it affects the true content length, this paper should be considered for desk rejection on the grounds of violating submission format and page-limit policies.

**Questions:**

See weaknesses.

---

> ### Author Response · Authors · 2025-11-20
> **Reponse to Reviewer 8qjz**
>
> > *“The submission does not comply with the ICLR formatting requirements…”*
>
> Thank you for pointing this out. We have corrected the margin and layout settings to fully comply with the official ICLR template in the latest revision. We would be grateful if you would review the current manuscript. We welcome any suggestions you may have for its improvement. Thank you.

---

### Official Review · Reviewer_4KCc · 2025-11-06

**Soundness:** 1
**Presentation:** 2
**Contribution:** 2
**Rating:** 2
**Confidence:** 4

**Summary:**

The authors proposed a novel speculative decoding method by reusing previous rollouts in RLVR as draft proposals. In this way, the RLVR training is free from individual draft model and utilize the past rollout trajectories. The authors introduce lenience to finegrainedly control the level of acceptance. As a result, nearly doubled speedups are seen on different benchmarks.

**Strengths:**

- The SPEC-RL is free from deployment of a seperate draft model.
- It exhibited significant token savings for superior performance on benchmark.

**Weaknesses:**

1. The most concerning issue of this paper is that SPEC-RL is **not algorithmically equivalent** to vanilla training. Standard speculative decoding is already not lossless acceleration under non-greedy sampling. Not to mention the SPEC-RL uses old trajectories rather than draft outputs.
2. The accept length is generally over hundreds. This is very alerting because it is far from most speculative decoding work whose acceptance length is generally not more than 5.
3. The introduced lenience variable is empirically set. This variable is severly impacting training as the authors analyzed. The justification for this variable and its value lacks theoretical proof.
4. The RLVR training highly dependent on sample variability (Especially for grouped rollout in GRPO and Reinforce++). In contrast, reusing in SPEC-RL significantly harms sampling diversity due to shared prompts.
5. Complete training trajectories and number of traning steps are not shown. It is hard to confirm convergence.

**Questions:**

1. The variable control is not clear. The description at line 666 said "we control for the total amount of rollout data:". But how? Is this refer to same amount of output token (incl. draft acceptance) or same number of prompts?
2. Assuming the number of trajectories put into the model is identical for each entry in Tab. 1, to my understanding, SPEC-RL is receving trajectories with "older" tokens. In this way, how could it generally show superior performance over baseline? (In contrast, it is intuitively acceptable if SPEC-RL can achieve similar performance in shorter time.)
3. How was the importance sampling done? Are draft using the log probs from previous round or newly computed ones?
4. The dataset access order is also confusing. To make sure old trajectories can be used, the following step must use the prompt in previous step. Considering the grouped rollout (N>1), the prompt accessed in SPEC-RL is more heavily exploited than vanilla training. Is this carefully controlled?
5. Considering that canonical speculative decoding strategies like EAGLE or Medusa can already attain a good gain on RLVR training, is there convincing reasons SPEC-RL is superior?

---

> ### Author Response · Authors · 2025-11-20
> **Reponse to Reviewer 4KCc**
>
> **(1) Difference from canonical speculative decoding**
>
> > *“Standard speculative decoding is already not lossless acceleration under non-greedy sampling.”*
>
> Our method fundamentally differs from canonical speculative decoding (e.g., multi-round *draft-and-verify*). Traditional speculative decoding performs several short draft rounds (typically 4–8 tokens per round). In contrast, **SPEC-RL treats the old policy itself as the draft generator** and the current policy as the verifier. Each draft is a *complete trajectory* rather than a short segment, and only one speculative round is performed. Since adjacent epochs in RLVR have highly similar policies, a large portion of the tokens remain acceptable even with a single round. **This effectively upgrades speculative decoding from short, multi-round token drafts to a single-round, trajectory-level reuse, yielding much longer accepted prefixes and higher acceleration. It also simplifies engineering: no extra draft model needs to be loaded—SPEC-RL only reads cached outputs from the previous policy—making integration into RLVR pipelines lightweight and straightforward.**
>
> **(2) Why reuse length cannot be compared to ordinary speculative decoding**
>
> > *“The accept length is generally over hundreds. This is very alerting because it is far from most speculative decoding work whose acceptance length is generally not more than 5.”*
>
> In canonical speculative decoding, the draft model is *smaller* and distributionally different from the target model, hence multiple verify rounds are required and the accepted span is short. In SPEC-RL, both draft and target are the *same model across adjacent epochs*. Therefore, reuse occurs along the entire old-policy trajectory and yields long acceptance spans by design. It is thus inappropriate to assess our reuse length by the “multi-round acceptance” standard used in traditional speculative decoding.
>
> **(3) Justification of the lenience parameter**
>
> > *“The introduced lenience variable is empirically set… lacks theoretical proof.”*
>
> The *lenience* parameter is not introduced as a new theoretical construct in our work. As noted in the main paper (lines **183–184**), we follow the design used in prior speculative-decoding work, where this parameter was originally proposed to control acceptance aggressiveness and improve reuse efficiency. Both that prior work and our experiments consistently show that tuning this parameter is an effective and stable way to balance reuse length and verification quality. Our use of *lenience* therefore follows an established design choice rather than introducing a new mechanism.
>
> **(4) Effect on sampling diversity**
>
> > *“Reusing in SPEC-RL significantly harms sampling diversity due to shared prompts.”*
>
> Empirically, we do **not** observe a reduction in rollout diversity when comparing **GRPO vs. GRPO + SPEC-RL** at the *same training step*. In the revision, we have added a new analysis in **Appendix A.9** (marked in blue in the updated manuscript), where we evaluate rollout diversity using **Distinct-1** [1] and **Self-BLEU** [2]. The results show that SPEC-RL maintains comparable—or slightly higher—diversity across training. SPEC-RL only changes how the prefix is computed at that step by reusing the previous policy’s rollout; it does **not** reduce the number of trajectories generated at the current step, nor does it constrain how the suffix tokens are sampled. As a result, when we take any given step and compare the rollout sets produced by default GRPO and by SPEC-RL under identical configurations, their sampling diversity remains highly similar. This is further reflected in the final benchmark results, where SPEC-RL consistently matches or exceeds baseline performance, indicating that diversity is preserved in practice.
>
> [1] *A Diversity-Promoting Objective Function for Neural Conversation Models.* NAACL 2016. https://aclanthology.org/N16-1014/
>
> [2] *Texygen: A Benchmarking Platform for Text Generation Models.* SIGIR 2018. https://dl.acm.org/doi/10.1145/3209978.3210080

---

> ### Author Response · Authors · 2025-11-20
> **Reponse to Reviewer 4KCc**
>
> **(5) Training steps and convergence**
>
> > *“Complete training trajectories and number of training steps are not shown.”*
>
> As reported in the Appendix, we originally trained **90 steps on DeepMath-6K** and **100 steps on SimpleRL-8K**. To directly address the reviewer’s concern, we have now included **substantially longer training trajectories**—up to **500 steps**—comparing baseline GRPO with GRPO + SPEC-RL on DeepMath-6K. We also updated our evaluation to more stable metrics—**AMC23-AVG@16** and **AIME24/25-AVG@32**—to better reflect the model’s reasoning ability across configurations. The full results are shown in the table below.
>
> Throughout the whole 500-step horizon, SPEC-RL consistently delivers **≈2× rollout speedup** and achieves **comparable or higher** performance than baseline GRPO at most checkpoints. This indicates that SPEC-RL does **not** hinder convergence, and its gains are not limited to early training but remain stable even when training is substantially prolonged.
>
> |Algorithm|Step|Tokens(M)|Speedup|AMC23 AVG@16|AIME24 AVG@32|AIME25 AVG@32|GSM8K|MATH500|Minerva|Olympiad|MMLU-STEM|IFEval|AVG|
> |--|--|--|--|--|--|--|--|--|--|--|--|--|--|
> |GRPO|100|616.5|1.00×|37.8|7.1|4.1|82.9|66.0|26.1|28.4|58.6|24.0|37.2|
> |GRPO|200|1220.6|1.00×|35.2|8.4|4.6|83.6|65.8|28.7|29.3|62.7|29.6|38.7|
> |GRPO|300|1816.3|1.00×|36.6|8.2|4.4|82.8|65.4|28.3|29.2|64.9|29.2|38.8|
> |GRPO|400|2401.7|1.00×|36.4|7.1|4.1|83.9|66.8|27.2|30.8|65.7|26.2|38.7|
> |GRPO|500|2970.0|1.00×|39.7|7.6|4.4|84.2|67.0|26.8|31.1|67.3|28.5|39.6|
> |with SPEC-RL|100|199.6|2.33×|38.3|9.3|4.2|83.9|66.4|27.2|29.3|61.9|27.9|38.7|
> |with SPEC-RL|200|365.6|2.43×|43.4|11.7|6.1|84.8|69.2|27.2|33.3|66.5|28.3|41.2|
> |with SPEC-RL|300|582.3|2.28×|41.6|13.9|6.6|84.3|67.2|28.3|32.1|65.9|27.2|40.8|
> |with SPEC-RL|400|840.1|2.12×|46.4|14.5|6.0|84.7|68.8|29.8|33.3|69.6|26.8|42.2|
> |with SPEC-RL|500|1118.1|1.99×|44.1|13.4|4.0|84.2|69.2|29.0|31.7|67.6|29.2|41.4|
>
> **(6) Clarification of variable control**
>
> > *“The variable control is not clear… ‘we control for the total amount of rollout data’—but how?”*
>
> Apologies for the ambiguity. DAPO adopts dynamic sampling, where the generation batch size is typically set to be several times larger than the training batch size. For example, with a training batch size of **1,024**, a generation batch size of **3,072** corresponds to roughly **3×** the standard rollout budget. To ensure a fair and controlled comparison with GRPO and PPO, we therefore use the **same generation batch size across methods**, so that all approaches operate under an equivalent rollout budget.
>
> **(7) Influence on entropy and reward**
>
> As shown in **Figure 6a** and **Figure 5a**, SPEC-RL has **minimal influence** on both entropy and reward throughout training. This is because all reused prefixes are first **verified by the current policy**, so they never function as outdated training signals. After verification, the accepted prefixes are treated exactly like freshly generated tokens and are re-evaluated under the current policy during log-prob computation, which preserves the algorithm’s **on-policy nature** and ensures that reuse affects efficiency but not the correctness of the optimization process.
>
> **(8) Importance-sampling computation**
>
> > *“How was the importance sampling done? Are drafts using log probs from previous round or newly computed ones?”*
>
> As shown in Eq. 4, we must compute the log probabilities for both the old and current policies. In our implementation, we store the log probabilities from previous rounds to avoid redundant computation. The full code implementing this behavior is provided in the supplementary material under  **`Supplementary-material/train/verl/trainer/ppo/ray_trainer.py` (lines 637–680).**
>
> **(9) Dataset access order**
>
> > *“The dataset access order is also confusing… Is this carefully controlled?”*
>
> Please note that SPEC-RL reuses rollouts from the **previous epoch**, not from the previous step. This is explicitly illustrated in **Fig. 3 (epoch t vs. epoch t–1)** and documented in multiple locations in the paper (Lines 166–167, 251–252, 477–478, and Figs. 12–15). To access them, we maintain a cache that stores all rollouts together with their log probabilities (Lines 218–219). Our data access order remains consistent with all baselines to ensure a fair comparison.

---

> ### Author Response · Authors · 2025-11-20
> **Reponse to Reviewer 4KCc**
>
> **(10) Advantage over canonical speculative decoding**
>
> > *“Considering that canonical speculative decoding strategies like EAGLE or Medusa can already attain a good gain on RLVR training, is there convincing reason SPEC-RL is superior?”*
>
> To the best of our knowledge, no prior work has successfully applied canonical speculative decoding to RLVR training. In fact, such an application faces two fundamental issues:
>
> 1. Deploying a small draft model efficiently poses significant challenges for current RL infra (e.g., verl), requiring substantial engineering effort.
> 2. During RLVR training, the target model is continuously updated while the draft model remains static. This causes a growing distribution gap over time, which hurts the efficacy of the rollouts.
>
> SPEC-RL avoids both limitations by using the previous policy as the draft model. This approach is much easier to implement and provides substantial rollout acceleration without degrading accuracy.

---

### Note · Authors · 2026-01-02

I have read and agree with the venue's withdrawal policy on behalf of myself and my co-authors.